# Single cell transcriptome atlas of the *Drosophila* larval brain

**Clarisse Brunet Avalos[1], G Larisa Maier[1], Rémy Bruggmann[2], Simon G Sprecher[1]***

[1]Department of Biology, University of Fribourg, Fribourg, Switzerland; [2]Interfaculty Bioinformatics Unit, University of Bern, Bern, Switzerland

**Abstract** Cell diversity of the brain and how it is affected by starvation, remains largely unknown. Here, we introduce a single cell transcriptome atlas of the entire *Drosophila* first instar larval brain. We first assigned cell-type identity based on known marker genes, distinguishing five major groups: neural progenitors, differentiated neurons, glia, undifferentiated neurons and non-neural cells. All major classes were further subdivided into multiple subtypes, revealing biological features of various cell-types. We further assessed transcriptional changes in response to starvation at the single-cell level. While after starvation the composition of the brain remains unaffected, transcriptional profile of several cell clusters changed. Intriguingly, different cell-types show very distinct responses to starvation, suggesting the presence of cell-specific programs for nutrition availability. Establishing a single-cell transcriptome atlas of the larval brain provides a powerful tool to explore cell diversity and assess genetic profiles from developmental, functional and behavioral perspectives.

*For correspondence:
simon.sprecher@unifr.ch

**Competing interests:** The authors declare that no competing interests exist.

## Introduction

The brain, as the central organ of the nervous system, shows high complexity and diversity of cell-types. Numerous tasks need to be synchronously orchestrated, singular areas are committed to specific functions and ultimately cause or modulate an array of complex behaviors. The first instar *Drosophila melanogaster* larval central nervous system (CNS) is composed of an estimated 10,000 cells (*Scott et al., 2001*). Only 2000 of these cells populate the two larval cerebral lobes, the remaining cells are distributed among segmental ganglia of the ventral nerve cord (VNC). The cells populating the larval brain develop from neuroblasts delaminated from the procephalic neurectoderm, during early embryonic stages. At the end of embryogenesis neurons are fully differentiated and form the functional neural circuits of the larval brain, while neuroblasts enter a mitotic quiescence phase and are only reactivated at the end of the first larval instar. Neuroblasts will re-enter proliferation and generate different cell-types that form the adult brain. During these steps, nutrient accessibility plays a key role. It has been previously described that some glial cells, in close proximity to the neuroblast populations, release insulin-like peptides upon nutrient-sensing. This signal is later incorporated by neuroblasts through the InR/PI3K/TORC1 pathway, to ultimately induce reactivation and exit from quiescence (*Chell and Brand, 2010*; *Sousa-Nunes et al., 2011*). Surprisingly, at late-larval stages, NPCs seem to be able to proliferate even in aversive feeding conditions, independently of the InR/PI3K/TORC1 signaling pathway (*Cheng et al., 2011*). Thus, the lack of nutriments may affect the molecular profile of the specified cell-types, consequently modifying the cellular state and composition of the larval brain. Therefore, identifying genetic responses during brain development in normal feeding condition versus starvation may allow a better and more complete understanding of the processes regulated by the intake of nutrients at early life stages.

The simplicity in cell number, in comparison to other animals, makes *Drosophila* larva an ideal candidate to establish a comprehensive catalogue of brain cell-types based on morphologies, developmental trajectories and synaptic connections between each other. Recently, the advent of single-

cell RNA sequencing (scRNA-seq) analysis further provides a high-resolution transcriptomic approach to decipher the molecular footprint at cellular resolution, as done to reveal the cell atlas of the adult brain (*Croset et al., 2018*; *Davie et al., 2018*; *Konstantinides et al., 2018*).

Here, we used a single-cell transcriptomic approach to establish a molecular cell atlas of the first instar larval brain. In this way, we identified five major cell-types: neural progenitor cells, neurons, glial cells, undifferentiated neurons and non-neural cell-types. Among differentiated cells we characterized expression and co-expression of distinct types of neurotransmitters, neuromodulators and neuropeptides, as well as distinct types of glial cells. We differentiated three major classes of neural progenitor cells (NPCs): neuroblasts, optic lobe precursors as well as mushroom body neuroblasts. We further analyzed non-neural cells from tissues that are anatomically closely associated with the brain, such as the prothoracic gland, the ring gland, *corpora allata*, fat body and muscles. Moreover, we observed the presence of presumptive undifferentiated neurons. For the various classes of brain cell-types and subtypes, our work further extended the list of previously described marker genes, which in turn may be used for developmental or functional studies. Finally, we described single-cell changes at the transcriptional level in the larval brain driven by starvation. We identified different cell clusters that show strong responses in gene expression upon starvation. Expectedly, the most striking differences were observed in NPCs, glial cells and undifferentiated cells. Interestingly, the response to starvation differed between distinct clusters, suggesting the presence of cell specific programs to nutritional changes.

## Results

### Single-Cell RNA-Seq of the *Drosophila* larval brain reveals different categories of characteristic cell-types

To investigate cellular and molecular diversity in the *Drosophila* larval brain, we performed single-cell transcriptomics analysis by applying 10X Genomics technology. In short, we dissected the CNS of late first instar larvae and separated the brain lobes from the VNC to enrich for cells populating the larval brain (*Figure 1A*). We obtained a cell atlas of the first instar larval brain with a total of 9353 cells and a median of 1658 genes per cell. The atlas is composed by cells from two different feeding conditions, normal feeding and 4 hr starvation prior brain dissection. Each experimental condition was treated separately and the libraries obtained were sequenced, aggregated and processed according to the standard pipeline for 10X single cell gene expression. In summary, the normal dataset comprised 4708 cells with a median of 1434 genes per cell and 836,393 mean reads per cell. While the starvation dataset comprised 4645 cells with a median of 1962 genes per cell and 545,512 mean reads per cell (*Supplementary file 1*).

To characterize the cellular population of the larval brain under normal feeding conditions, we further analyze the 4708 cells belonging to the normal experimental condition. These cells were filtered, scaled and normalized using Seurat R package (*Butler et al., 2018*; *Stuart et al., 2018*). The resulting 4349 cells, with 12,942 genes detected, were later clustered generating 29 initial cell-clusters that were subsequently visually represented using a novel learning technique for dimensional reduction, Uniform Manifold Approximation and Projection (UMAP) (*McInnes et al., 2018*) (*Figure 1—figure supplement 1A*). Clusters were manually annotated based on previously identified marker genes for specific cell-types, typically assessing the top 10 differentially expressed genes across clusters (*Figure 1—source data 1*). Among the 29 clusters, we distinguished five main cell type categories: differentiated neurons, NPCs, glial cells, undifferentiated neurons and a group of non-neural cells (*Figure 1B*). Differentiated neurons could be defined by the expression of the pan-neuronal marker *embryonic lethal abnormal vision* (*elav*), presynaptic genes or widely expressed neurotransmitters, NPCs by *Notch* (*N)* or *deadpan* (*dpn*) expression, and glial cells by the expression of *reversed polarity* (*repo*) (*Figure 1C,D*, *Figure 1—figure supplement 1B*). Some of these cells displayed particular overlaps between their expression profiles, enabling a more simplified version of the UMAP plot with only 15 clusters (*Figure 1—figure supplement 1A*). Within these clusters we distinguished mushroom body Kenyon Cells (KC), neurons expressing distinct peptides and neurotransmitter-releasing neurons, among the differentiated cells. While in the group of non-neural cells, we could differentiate cells coming from the ring gland, *corpora allata*, imaginal discs, hemolymph, lymph gland and fat tissue (*Figure 1—figure supplement 2A*). The identification of these last

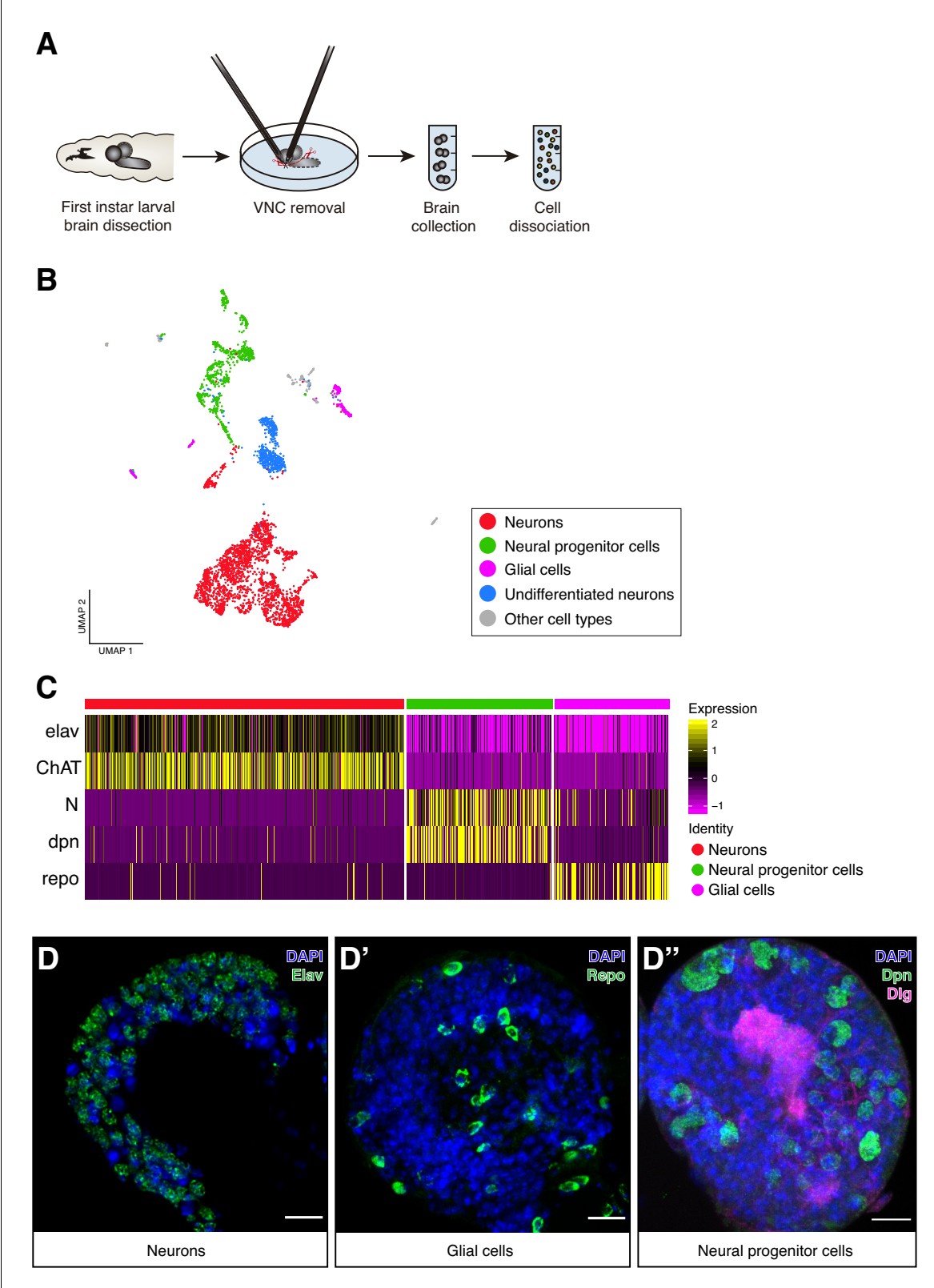

**Figure 1.** Cellular composition of the Larval brain by Single-Cell RNA-Seq. (A) Experimental procedure. *Drosophila* late first instar larval brains/ventral nerve cords (VNC) were dissected and the VNCs were removed. Brains were collected and dissociated into a suspension of single cells. (B) Cell atlas of the larval brain reveals five main cell-types: neurons, NPCs, glial cells, UNs and other cell-types, represented in a Seurat UMAP plot. Groups are color coded. (C) Cell-types are recognized based on the expression of previously characterized marker-genes. A simplified heatmap illustrates this process: a

*Figure 1 continued on next page*

Figure 1 continued

subgroup of cholinergic neurons expresses the pan-neuronal marker *elav* and *ChAT*, a protein that catalyzes the biosynthesis of the neurotransmitter acetylcholine; NPCs are recognized by the expression of *Notch* and *dpn* and glial cells by *repo* expression. (**D–D''**) Validation of the markers used to identify the different cell-types within the larval brain by immunostainings. The images display one lobe of the larval brain. Nuclei were labeled with DAPI and cellular borders with *Dlg* (*disc large*). Scale bar: 10 μm.

The online version of this article includes the following source data and figure supplement(s) for figure 1:

**Source data 1.** Clusters properties.
**Figure supplement 1.** Cell-type catalogue of the Larval brain.
**Figure supplement 2.** Non-neural cell-types identified in the dataset.

category demonstrates that during the dissection protocol, while the VNC was removed, different organs remained at least partly attached to the brain. Interestingly these cells clustered separately, further highlighting the power of the methodology.

## Mapping neural progenitor cell diversity reveals presence of characteristic neurogenic cell-types

Several clusters showed prominent expression of key genes involved in different steps of neurogenesis. We could identify a population of *Notch* (*N*) positive cells, divided into several clusters closely located in our dimensionality reduction representation. To further confirm the identity of these cells, we analyzed the expression of known NPCs markers. In addition to *N*, these cells were expressing *dpn*, *asense* (*ase*), *klumpfuss* (*klu*), *pointed* (*pnt*) and *prospero* (*pros*) (*Figure 2—figure supplement 1A*). During embryonic development, neuroblasts are known to follow a temporally sequential expression of transcription factors, also known as temporal cascade: *Hunchback* (Hb), *Kruppel* (*Kr*), *Pdm1/Pdm2* (*Pdm*), *Castor* (*Cas*) and *Grainy head* (*grh*) (*Allan and Thor, 2015*). The early cascade gene *Kr* was observed to be expressed in differentiated cells, while the late gene *grh* was almost exclusively expressed in cells from the neurogenic clusters (*Figure 2A*).

To further resolve NPCs, we sub-clustered all the neurogenic populations, obtaining a new representation for these cells now grouped in 11 clusters (*Figure 2B*). Next, we examined the top differentially expressed genes among each group and we observed a high degree of similarity in their transcriptional programs. Interestingly, we detected a high number of cells expressing long non-coding RNAs (lncRNAs); some of them with reported functions, as it was the case of *cherub* (*lncRNA: CR43283*), known to be involved in tumorigenesis and in normal brain development (*Landskron et al., 2018*; *Malin and Desplan, 2018*), but also lncRNAs with unknown functions, like *lncRNA:CR30009* (*Figure 2C*). In parallel, we observed expression of genes reported to be actively transcribed in the optic lobe neuroepithelium, such as *Ocho* (*Ocho*), *Twin of m4* (*Tom)* and *Bearded* (*Brd*) (*Egger et al., 2010*) in a subpopulation of cells; therefore, this group of cells was annotated as the optic lobe epithelium (OLE) (*Figure 2D*). In addition to these genes, we identified basic helix-loop-helix (bHLH) transcription factors, such as *E(spl)m8-HLH* and *E(spl)mgamma-HLH* (*Figure 2D*). The remaining clusters showed different combinations of the following genes, all reported to be involved in neurogenesis: *deadpan* (*dpn*), *klumpfuss* (*klu*), *asense* (*ase*), *earmuff* (*erm*), *pointed* (*pnt*) and *prospero* (*pros*), among others (*Figure 2—figure supplement 1B*).

Knowing that NPCs upon embryogenesis, beside mushroom body and lateral neuroblasts, enter a quiescence state and that later, upon nutritional-signals reactivate (*Egger et al., 2008*); we evaluated cell-cycle genes expression in the NPCs population. We observed that the different NPCs displayed distinct cell-cycle marker genes, resulting in a heterogenic population of cells. We found a larger number of cells in G1 and G2/M phases, in comparison to S phase (*Figure 2E*). These results further support recent observations, where G2 phase was shown to be an entry point to quiescence, alongside G0 phase (*Otsuki and Brand, 2018*). Surprisingly, a relevant number of NPCs expressing marker genes for S phase were observed, indicating that these cells may be replicating their genomic content in order to proliferate. Therefore, we performed EdU incorporation experiments, together with *dpn* immunostaining, and identified approximately 30% of *dpn* positive EdU positive cells, from the total of *dpn* positive cells (*Figure 2F*); suggesting that at 16 hr ALH (after larval hatching) some NPCs were already exiting their dormant state and reassuming proliferation.

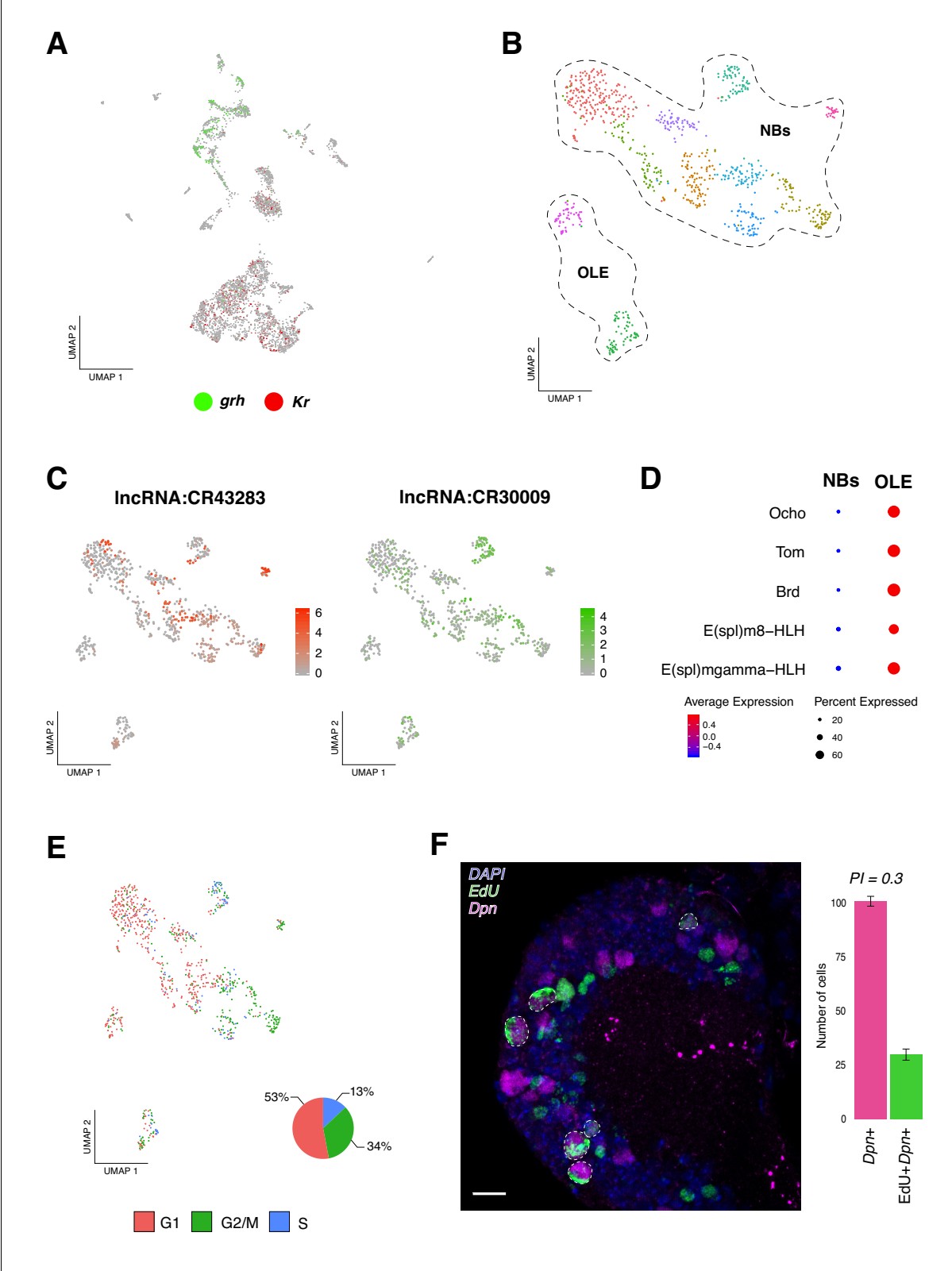

**Figure 2.** Identification of neural progenitor cell populations. (**A**) Expression-pattern of genes from the temporal cascade distinguish population of cells. Two dimensions UMAP plot labeling genes from the temporal cascade. *Grh, grainy head* in green and *Kr, Kruppel* in red. *Grh* is broadly expressed in the recently born neuroprogenitors, while *Kr* is expressed in mature cells. (**B**) Re-analysis of the neurogenic populations by sub-clustering the original NPCs population, plotted in a Seurat UMAP plot, identifies 11 sub-populations of cells. NPCs are further divided into neuroblasts and optic

*Figure 2 continued on next page*

*Figure 2 continued*

lobe epithelium (OLE). (**C**) lncRNAs are present in NPCs expression profile. Color coded UMAP plot showing the abundance of long non-coding RNAs (lncRNAs). The scale represents gene expression levels. (**D**) OLE can be distinguished from the remaining neuroblasts based on the expression of characterized marker genes. Dot size corresponds to the percentage of cells expressing a particular gene, while color intensity represents gene expression levels. (**E**) Cell-cycle scores can be estimated transcriptionally. UMAP plot showing the different NPCs color coded based on cell-cycle phases. The pie graph illustrates the percentage of NPCs in each phase, G1, G2M and S. (**F**) EdU incorporation and NPCs proliferation. Immunostaining of the larval brain illustrating *Dpn+* proliferating NPCs at 16 hr ALH. Proliferation index (PI) was calculated as the ratio between *Dpn+EdU+* NPCs and *Dpn+* NPCs. Error bars represent standard error. N = 5. Nuclei were labeled with DAPI. Scale bar: 20 μm.

The online version of this article includes the following figure supplement(s) for figure 2:

**Figure supplement 1.** Marker genes for the identification of neural progenitor cell populations.

**Figure supplement 2.** Identifying undifferentiated neurons in the larval brain.

In close proximity to the neurogenic population in the UMAP plot, we identified two clusters of cells that lack expression of neurogenic marker genes, but express neuronal markers such as *elav, Syt1 (Synaptotagmin 1)* and *nervana 3 (nrv3)*. However, these clusters lacked of marked expression of genes required for neurotransmitter biosynthesis. In addition, we found high expression of *head-case (hdc)* and *unkempt (unk)*, two genes implicated in controlling proper timing of neural differentiation (*Avet-Rochex et al., 2014*). Other genes enriched in these cells were *Thor, scylla (scyl), hikaru genki (hig)* and *Broad-Z3 (br)* (*Figure 2—figure supplement 2A*). Since these cells clustered in vicinity of NPCs, apart from the remaining differentiated neurons, their lack of neurotransmitter markers and their expression of differentiation genes, we labeled these cells as presumptive undifferentiated neurons (UNs): UNs 1 and UNs 2. However, further analyses and validations are required to establish their bona fide identity.

## Neuronal cell-types co-express different neuroactive molecules

We next analyzed the population of mature cells, more precisely those cells positive for synaptic markers: *synaptobrevin (nSyb)* and *synaptotagmin (Syt)* (*DiAntonio et al., 1993*), and evaluated the expression of genes involved in the release or synthesis of neurotransmitters: *glutamate (Glu)*, *acetylcholine (ACh)*, *gamma-aminobutyric acid (GABA)* and *monoamines*. We classified cells as glutamatergic, cholinergic, GABAergic and monoaminergic neurons; as they expressed: *vesicular glutamate transporter (VGlut)*, *vesicular acetylcholine transporter (VAChT)*, *Glutamic acid decarboxylase 1 (Gad1)* or *vesicular monoamine transporter (Vmat)*, respectively. *Glu* and *Ach* were found to be the most abundant neurotransmitters, as they were each expressed in 24% of the total number of neurons. GABAergic and monoaminergic neurons occupied the third and fourth place, with 17% and 10% of the total number of neurons, respectively. (*Figure 3A–B*).

In past years, the co-existence of more than one neurotransmitter in the same neuron has been extensively studied in mammals. A neuron could release multiple neurotransmitter types from the same synaptic vesicle, from different vesicles at the same synapse or even at different synaptic boutons (*Vaaga et al., 2014*). The recent publications on the *Drosophila* adult brain cell atlas re-opened this subject, by providing a list of co-expressed neuroactive molecules in the brain (*Croset et al., 2018*; *Davie et al., 2018*; *Nässel, 2018*). Therefore, we investigated the phenomenon of dual or multiple transmitter neurons in the larval brain. We analyzed all the possible combinations of the four very well-known marker genes for neurotransmitters: *VGlut; VAChT; Gad1* and *Vmat*. Remarkably, we found an overlap in all combinations, although the occurrence of these events represents a small fraction in the whole dataset. Dual-transmitter neurons were more frequent than triple-transmitter neurons, with the three highest overlap occurring between *VGlut* and *Vmat* (3%), followed by *VGlut* and *VAChT* (2%) and *VGlut* and *Gad1* (2%). Triple-transmitter expressing neurons only represent approximately 1% of the total of neurons (*Figure 3C–E*). Next, we validated these observations with immunostainings. We observed an overlapping between *VGlut* and *pale* (*ple*), a tyrosine hydroxylase involved in the synthesis of dopamine (*Neckameyer and Quinn, 1989*); as well as between *VGlut* and *Gad1* and *VAChT* and *Gad1* (*Figure 3—figure supplement 1*). Thereby, a subset of neurons actively co-expressed different neurotransmitters, showing that this phenomenon also occurs in the *Drosophila* larval brain.

Next, we further characterized the group of monoaminergic neurons, defined by the expression of *Vmat*. We analyzed the expression of specific markers: *dopa decarboxylase* (*Ddc*), *Tyramine β*

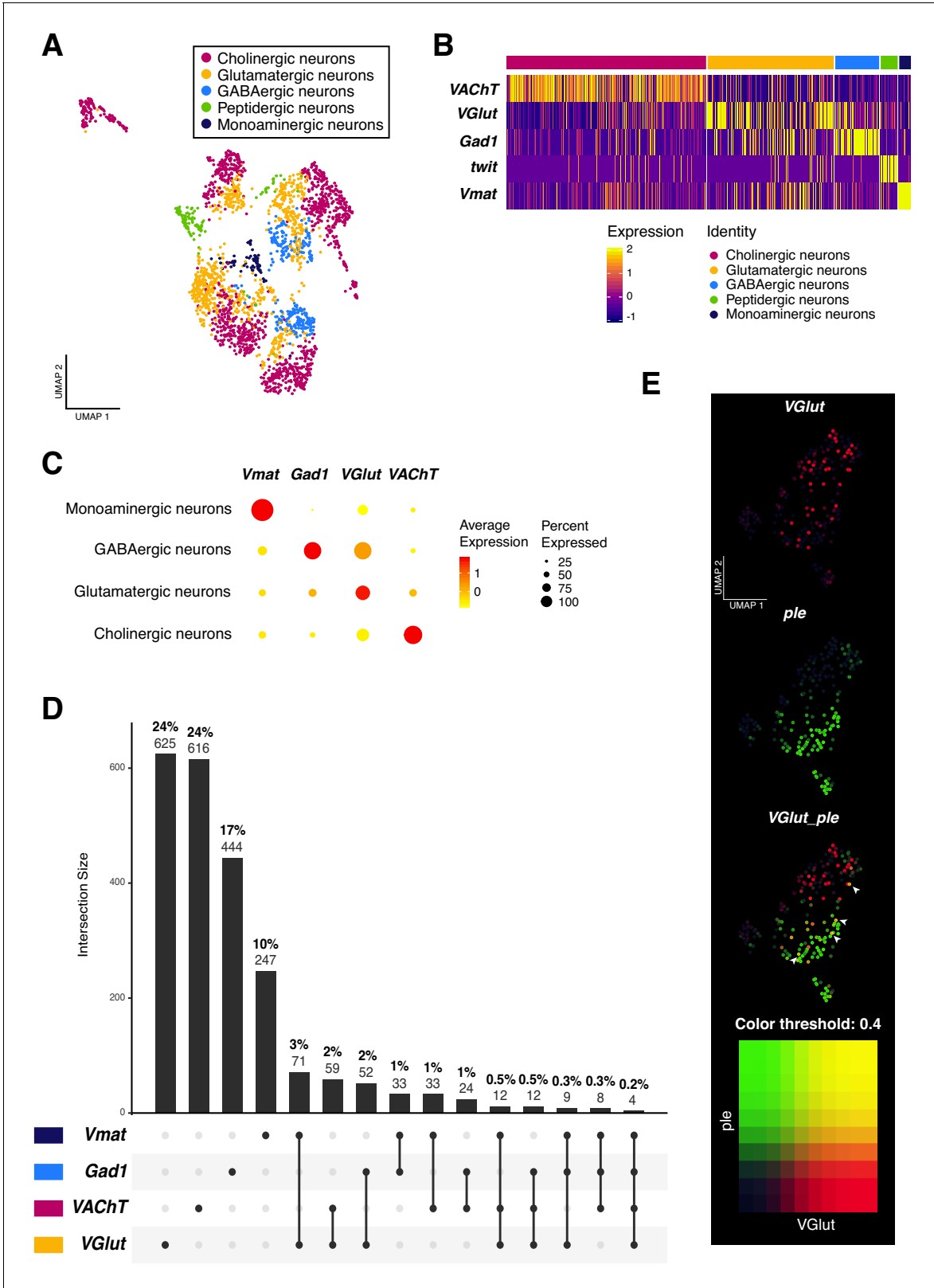

**Figure 3.** Neurotransmitter expression and co-expression in the larval neuronal population. (**A**) Neurons are classified based on the expression of neurotransmitters. Seurat UMAP plot showing the distribution of the five main neuronal cell-types. (**B**) Simplified heatmap, representing neuronal subpopulations. Genes displayed are the main markers analyzed to identify neuronal cell-identities. The x axis represents individual cells, each line corresponds to one neuron. Gene expression levels are color coded. (**C**) Neurotransmitters are co-expressed in a subset of neurons. Dot size

*Figure 3 continued on next page*

*Figure 3 continued*
corresponds to the percentage of cells expressing a particular gene, while color intensity represents gene expression levels. (D) UpSet plot
(**Conway et al., 2017**) illustrating the co-expression of neurotransmitters. Light and bold numbers represent number of cells and percentages of cells, respectively. The percentages were calculated based on positive cells for a particular neurotransmitter or a combination of them, in comparison to the total of neurons from the dataset. (E) Co-expression analysis based on the simultaneous expression of *ple* and *VGlut.* Blend UMAP plot showing only *ple* positive cells. Cells co-expressing *ple* and *VGlut* are shown in yellow. Threshold: 0,4.
The online version of this article includes the following figure supplement(s) for figure 3:

**Figure supplement 1.** Neurotransmitter co-expression in the larval brain.
**Figure supplement 2.** Monoaminergic neurons analysis.

hydroxylase (*Tbh*)/*Tyrosine decarboxylase 2* (*Tdc2*) (**Monastirioti et al., 1996**; **Roeder, 2005**) and *serotonin transporter* (*SerT*); in order to recognize dopaminergic (DA), octopaminergic(OA)/tyraminergic(TA) and serotonergic neurons, respectively. Interestingly, only *Ddc* and *ple* were expressed at high levels in monoaminergic neurons, therefore this cluster was labeled as dopaminergic neurons (***Figure 3—figure supplement 2A***). OA/TA and serotonergic neurons were displayed in different clusters, always in the presence of another neurotransmitter, further supporting our findings of dual expression of transmitter-molecules in the larval brain neurons (***Figure 3—figure supplement 2B***).

## Characterization of neuropeptide expression in the *Drosophila* larval brain

Among the population of mature cells, we found a cluster of cells that differentially expressed genes involved in protein synthesis, as well as the pan-neuronal marker *elav*. Since neuropeptide biogenesis requires the protein synthesis machinery, we wondered if these neurons were peptidergic. After analyzing the different genes being expressed, we observed that these cells contained different known neuropeptides, therefore we classified them as peptidergic neurons. The different neuropeptides followed different expression patterns; some were broadly expressed, while others were only expressed in a small number of cells.

To further characterize the peptidergic neurons (***Figure 4A***), we subdivided those cells that were positive for a specific neuropeptide and investigated their expression profiles. Knowing that the larval brain possesses five lateral neurons (LNs) per hemisphere, of which only four express the neuropeptide *Pigment-dispersing factor* (*Pdf*) (**Collins et al., 2012**; **Helfrich-Förster, 1997**), we decided to further characterize this cell-type. We distinguished the Pdf-LNs of the larval brain, and in addition to *Pdf*, these cells expressed core clock component genes: c*ryptochrome* (*cry*), *period* (*per*), *clock* (*clk*) and *timeless* (*tim*). In addition, PDF neurons also expressed the *Pigment-dispersing factor receptor* (*Pdfr*), suggesting a self-regulatory function for *Pdf* secretion. Moreover, genes implicated in circadian processes showed to be co-expressed with *Pdf*, as it was the case for the nuclear hormone receptors, *Hr51* and *Hr38*, and the transcription factor *vrille* (*vri*) (**Abruzzi et al., 2017**). Furthermore, we identified novel marker genes for this particular cell-type, some of them with described functions, such as the *Cyclic-AMP response element binding protein A* (*CrebA*), *Matrix metalloproteinase 2* (*Mmp2*), *Allatostatin C receptor 2* (*AstC-R2*) and the transcription factor *stripe* (*sr*); as well as other uncategorized genes, such as *CG11221*, *CG44153*, *CG43902* and *CG12541* (***Figure 4B***). In close vicinity with these cells, another population of neuropeptide-producing cells is involved in the secretion of the *Prothoracicotropic hormone* (*Ptth*). We mapped these cells to our dataset and we found that this gene was not being transcribed at high levels, compared to other neuropeptides, but it was indeed present in the larval brain (***Figure 4C***).

Following the previous analyses, we then focused on another neurosecretory cell-type, the insulin producing cells (IPCs). These cells are responsible for the synthesis of 3 main *insulin like peptides* (Ilps): *Ilp2*, *Ilp3* and *Ilp5* (**Nässel and Vanden Broeck, 2016**). We found a small set of Ilp positive cells, therefore we classified them as IPCs (***Figure 4D***). The secretion of these neuropeptides is thought to be regulated through nutrient sensing in cells located in the ring gland, more precisely in the cells releasing *Adipokinetic hormone* (*Akh*). Consequently, we decided to study the expression of *Akh* in cells considered to be non-neural cell-types, and we observed that *Akh* was present in the thought-to-be the prothoracic gland and not in the larval brain (***Figure 4C***). Additionally, some neuropeptides are known to require the peptidase *amontillado* (*amon*) in order to become bioactive,

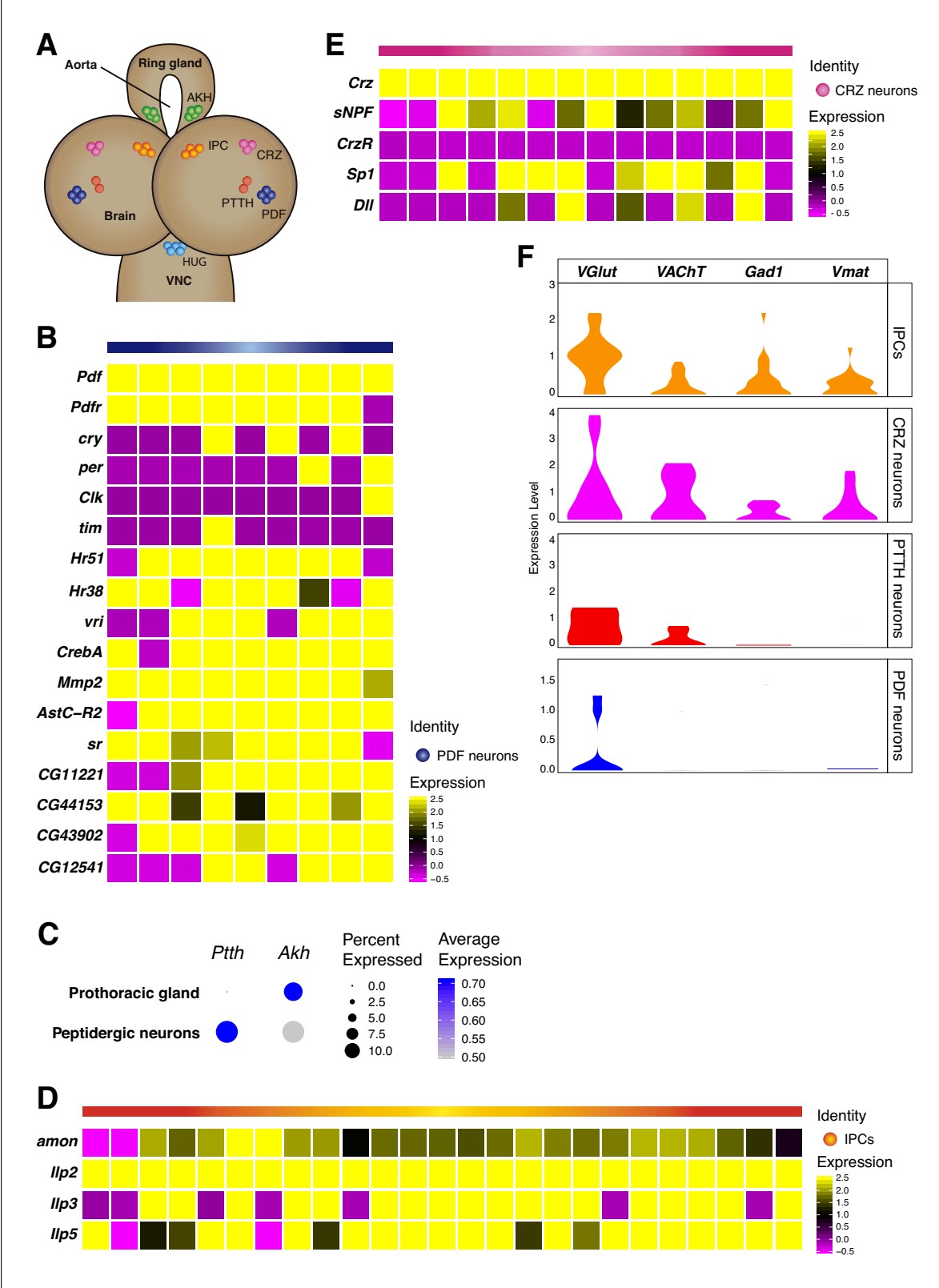

**Figure 4.** Neurosecretory cells distribution in the larval brain. (**A**) Schematic representation of some neurosecretory cells present in the larval brain and the ring gland. VNC, ventral nerve cord. (**B, D, E**) Heatmaps showing gene expression profiles of PDF neurons, IPCs and CRZ neurons. In each case, only *Pdf*, *Ilp2* and *Crz* highly positive cells were considered, respectively. White vertical lines separate individual cells, each column represents one cell. Gene expression levels are color coded. (**C**) Dot plot showing the presence of *PTTH* and *Akh* neurons, in peptidergic neurons and prothoracic gland,

*Figure 4 continued*

respectively. Dot size represents percentage of cells expressing a particular gene, while color intensity represents gene expression levels. (F) Neurosecretory cells express one or multiple neurotransmitters. Violin plot illustrating co-expression of neurotransmitters and neuropeptides. Violin plots represent the probability density of the data integrated with kernel density estimation. Wider sections of the violin plots represent a higher probability of cells with the indicated gene expression level, while skinnier sections represent lower probabilities.

we checked if this was also the case for the Ilps and noticed that *amon* mRNA levels were significantly high in the IPCs found in the larval brain (*Figure 4D*).

Next, we identified a set of neurons involved in the release of *Corazonin* (*Crz*), the CRZ neurons. However, in contrast to Pdf-expressing neurons, we did not observe an overlap between *Crz* and *Corazonin receptor* (*CrzR*) in these cells. Interestingly, we found different transcription factors being co-expressed in the CRZ neurons, such as *Distal-less* (*Dll*) and *Sp1* (*Figure 4E*). In addition, we observed the *short neuropeptide F* (*sNPF*) being co-synthesized in *Crz* positive cells, as it has been previously described (*Johard et al., 2008*; *Lee et al., 2004*). Nevertheless, *sNPF* has been reported to exhibit a wide expression in the larval brain (*Nässel et al., 2008*), and in addition to its expression in *Crz* neurons, it was also expressed in the mushroom body Kenyon cells (*Figure 5B*).

Finally, we analyzed the expression of genes required for neurotransmitter synthesis and transport in peptidergic neurons. We observed that PDF neurons were mostly glutamatergic, while IPCs, PTTH and CRZ neurons seemed to be more promiscuous, co-releasing different neurotransmitters. These findings are in agreement with the notion that a single neuron could in principle make use of different types of molecules for intercellular communication (*Figure 4F*).

## Larval mushroom body characterization identifies three major cell-types

A well-studied structure in the *Drosophila* brain is the mushroom body (MB), a region where olfactory learning and memory take place (*Davis, 2011*). In the adult fly each MB consists of approximately 2000 neurons, called Kenyon cells (KCs), these cells are produced by four mushroom body neuroblasts (MBNBs) per hemisphere, that arise during embryogenesis and continue dividing until pupal stages (*Kunz et al., 2012*). The neurons populating the larval MB, are embryonic-born KCs (*Pauls et al., 2010*). We next addressed if we could identify KCs based on known marker genes for the adult MB.

We first sub-clustered the population identified as mushroom body, identifying three types of cells based on cell-cycle phases and marker genes: mushroom body Kenyon cells (MBKCs), mushroom body neuroblasts (MBNBs) and cells undergoing differentiation (MBUNs) (*Figure 5A*). MBKCs expressed characteristic MB receptors: *Dopamine 1-like receptor 1* (*Dop1R1*); *Dopamine 1-like receptor 2* (*Dop1R2*) and *Dopamine 2-like receptor* (*Dop2R*). In addition, these cells were enriched for *Fasciclin 2* (*Fas2*), short *neuropeptide F* (*sNPF*) and *Neprilysin 1* (*Nep1*), among other genes (*Figure 5B*). Remarkably, *portabella* (*prt*), an orphan vesicular transporter previously reported to be expressed in the mushroom body (*Brooks et al., 2011*), was enriched in the larval KCs (*Figure 5B*). We confirmed this finding by immunostaining, where we observed an accumulation of *prt* in the MBKCs, more precisely in the calyx, penduncle, and vertical and medial lobes (*Figure 5C*). As expected, MBKCs appeared to be cholinergic (*VAChT*).

On the other hand, MBNBs were found to be constantly proliferating, as their cell-cycle scores indicated (*Figure 5A*). Moreover, these cells expressed characteristic neuroprogenitor marker genes, such as *N*, *asense* (*ase*), *Delta* (*Dl*) and *tailless* (*tll*). In addition, *target of Poxn* (*tap*), a gene playing a key role during mushroom body development was also upregulated in this cell population (*Figure 5B*). Finally, MBUNs expressed the neuronal markers, *nSyb* and *VAChT*, but at lower levels in comparison to MBKCs, indicating that they are not yet fully differentiated. Therefore, the particular composition and expression profiles of the MB cells could explain its location in the UMAP plot, as they are found closer to the NPCs than to neuronal clusters.

## Characterization of glial cell-types populating the larval brain

A major cell-type of the brain are the glial cells. Different types have been described in the adult brain: cortex, surface, neuropil and astrocyte like glia (*Croset et al., 2018*; *Konstantinides et al., 2018*). To identify the glial cell populations in the larval brain, we selected the marker genes for

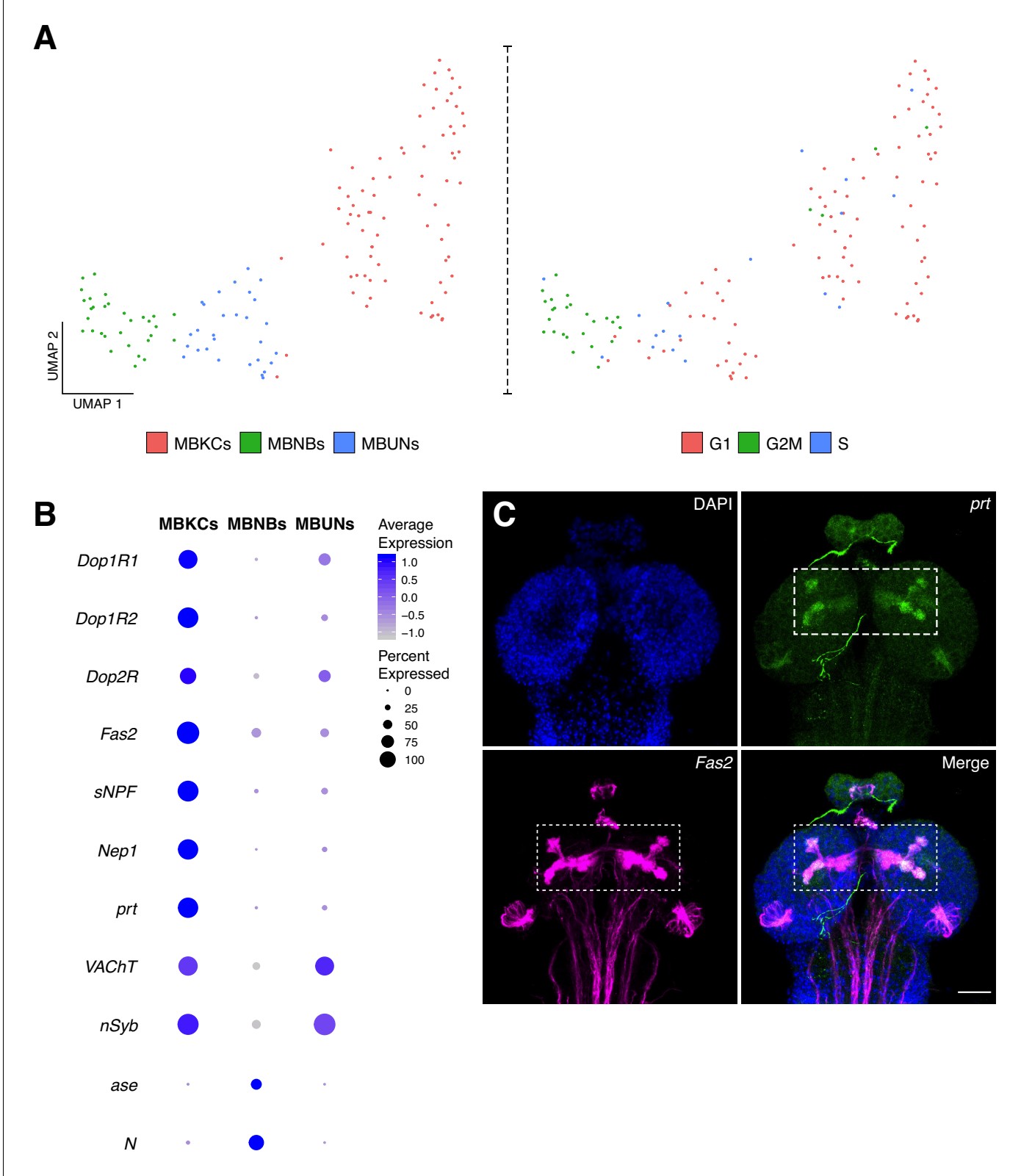

**Figure 5.** Characterization of the mushroom body cellular composition. (**A**) Identification of three main cell types among the mushroom body cluster of cells. Left panel: Re-clustering of the MB cluster led to three different cell populations: MBNBs, MB neuroblasts; MBUNs, MB undifferentiated neurons and MBKCs; MB Kenyon cells. Right panel: cell cycle analysis of the MB population further confirms previously mentioned cell-types. S, S phase; G1, G1 phase and G2M, G2/mitosis phases of the cell cycle. (**B**) The different cell types express distinct genes and validates their classification. Dot plot

*Figure 5 continued on next page*

*Figure 5 continued*

representation of the main markers found for each MB subpopulation. Dot size corresponds to percentage of cells expressing a particular gene, while color intensity represents gene expression levels. (**C**) Validation of marker genes found for the MBKCs. Immunostaining showing the distribution of *prt* in the larval brain MB. The MB was labeled with *Fas2* and nuclei with DAPI. In the merge condition, *prt* antibody colocalizes with *Fas2*, indicating their co-expression in the MB. Scale bar: 100 μm.

each cell-type and evaluated their expression. We observed three different clusters: astrocytes/neuropil glia, cortex/chiasm glia and surface glia, enriched for specific marker genes (*Figure 6A*).

Astrocytes like glia were identified based on expression of known marker genes: *astrocyte leucine-rich repeat molecule* (*alrm*), *Excitatory amino acid transporter 1* (*Eaat1*), *GABA transporter* (*Gat*), *Glutamine synthetase 2* (*Gs2*) and *wunen-2* (*wun2*) (*Huang et al., 2015*). Since we found another cluster of cells expressing *Eaat1* and *Gs2* in addition to *ebony* (*e*), and knowing that the last one is only expressed in neuropil glia, we annotated this cluster as astrocytes/neuropil glia. Then, cortex/chiasm glia was identified by the expression of *wrapper* (*wrapper*) (*Konstantinides et al., 2018*; *Noordermeer et al., 1998*) and *hoepel1* (hoe1). Lastly, surface glia was characterized by the following marker genes: *I'm not dead yet* (*Indy*) and *CG6126* (*Figure 6B*).

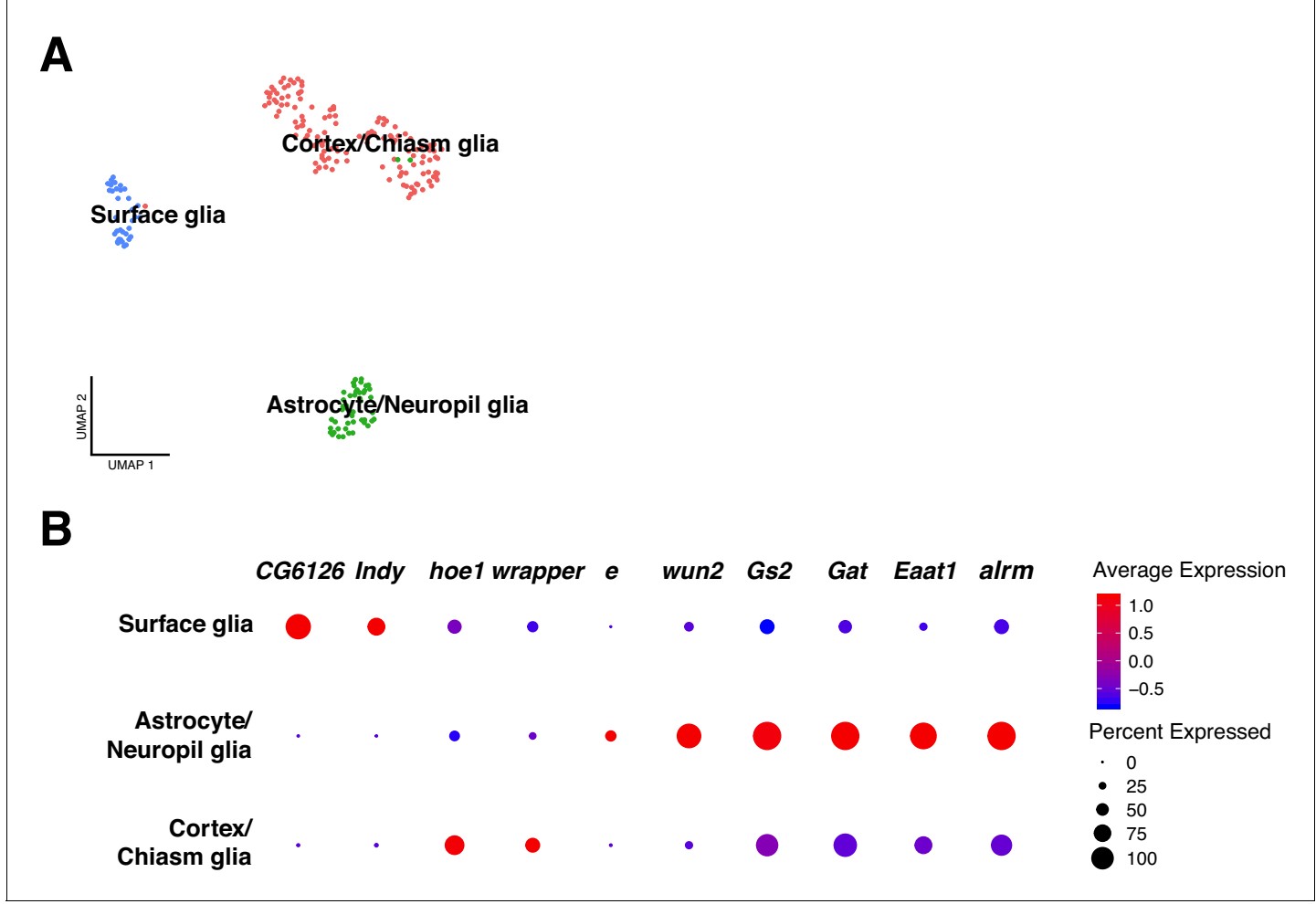

**Figure 6.** Classification of glial cell-types. (**A**) Analysis on the glial cell population revealed different glial cell types. After sub-clustering of the initially identified glial cells, three main clusters were identified: surface glia, astrocyte/neuropil glia and chiasm/cortex glia. (**B**) Each particular glial cell type possesses a particular expression profile. The differentially expressed genes are represented in a dot plot, where dot size corresponds to the percentage of cells expressing a particular gene and color to gene expression intensity levels. Red: high expression, blue: low expression.

## Starvation effects on larval brain cellular composition

Changes in the nutritional state of a brain may alter ongoing transcriptional programs and require additional or parallel ones. Here we focused on understanding how distinct cell-types respond to a specific brain state. Therefore, we set out to alter the availability of nutrients by starving first instar larvae for four hours. We observed that at this time point the initial larval population was reduced approximately two times, indicating the strong effect that starvation has on the survival rate (*Figure 7—figure supplement 1*). We then compared both larval brain cell atlases, normal versus starvation. We found that both datasets overlap for most clusters containing cells from each condition, with two clusters being present in only one condition. This suggests that after starvation, some cell-types were more strongly affected than others and that the changes in their transcriptional programs made them cluster separately (*Figure 7A*). Therefore, we proceeded to identify each cluster in order to understand where the effect of starvation had an impact on the larval brain cellular composition (*Figure 7B*).

First, we manually annotated the different cell clusters upon combining the two conditions, resulting in approximately the same cell-types largely described above. Among all the clusters, only two were missing after starvation: UNs 2 and Neurons X. To characterize these cell-types, we analyzed their expression profiles and we found that Neurons X exhibited a combination of different neurotransmitters that could be further subdivided into smaller clusters based on the expression of particular neurotransmitters (*Figure 7C*). Thus, a subset of cells of different neurotransmitter identities respond probably in a similar way so that they now form a joint cluster. Nevertheless, it is also possible that cells populating these clusters were stochastically lost during sample preparation and its posterior manipulation.

The second cluster missing upon starvation was UNs 2, which showed an overall similarity with the non-altered UNs 1 cluster. Both expressed genes involved in TORC1 signaling pathway: *Repressed by TOR* (*REPTOR*), *REPTOR-binding partner* (*REPTOR-BP*) and *Thor* (*Figure 7D*). In addition to these genes, we observed expression of *hdc* and *unk*, whose products form a complex and regulate cell cycle progression trough binding TORC1 components, in response to nutrient intake (*Li et al., 2019*). Furthermore, transcripts of genes involved in cell growth regulation were also detected, such as *charybde* (*chrb*) and *scylla* (*scyl*). Therefore, UNs 2 appear to be more sensitive to starvation conditions than the remaining cell types, explaining their absence after starvation (*Figure 7A*). Since *hdc* was found to be expressed in both NPCs and UNs, but only enriched in UNs, we quantified *hdc* expression in normal and starved brains. We observed a decreased number of *hdc* positive UNs upon starvation, as suggested in silico. In addition, these cells displayed lower levels of the pan-neuronal marker *elav*, in comparison to the surrounding mature neurons. These results suggest that UNs can be characterized by a combination of high levels of *hdc* expression an low levels of *elav* expression; and that in fact these group of cells were severely affected by the lack of nutrients (*Figure 7—figure supplement 2*).

In early larval stages neuroblasts are arrested in G0 or G2 phases of the cell-cycle (*Otsuki and Brand, 2018*) and a complex signaling mechanism between fat-body, glial cells and neuroblasts triggers their reactivation upon nutrient-sensing (*Chell and Brand, 2010*; *Sousa-Nunes et al., 2011*). Thus, we evaluated how cell-cycle marker genes expression was affected by the lack of nutrients. We observed that the above described cell-cycle heterogeneity was maintained upon starvation, but the proportion among cell cycle phases was altered. In addition, we noticed a decreased number of cells in G2M phases, and an increase in G1 and S phases; suggesting that upon starvation neuroblasts do not exit quiescence and therefore less cells undergo mitosis (*Figure 7—figure supplement 3*).

## Transcriptional responses to starvation

Once cell identities were assigned, we investigated changes at the transcriptional level. First we analyzed the expression of genes involved in fatty acid metabolism (*Zinke et al., 2002*). We found *withered* (*whd*), a carnitine O-palmitoyltransferase, being primarily upregulated in glial cells upon starvation. Moreover, other genes with key roles in fat catabolism were also upregulated, as was the case of the long-chain-fatty-acid-CoA ligase, *pudgy* (*pdgy*) and the *Lipase 4* (*Lip4*). Among genes that were downregulated after starvation we found the *fatty acid synthase 1* (*FASN1*) (*Figure 8A*).

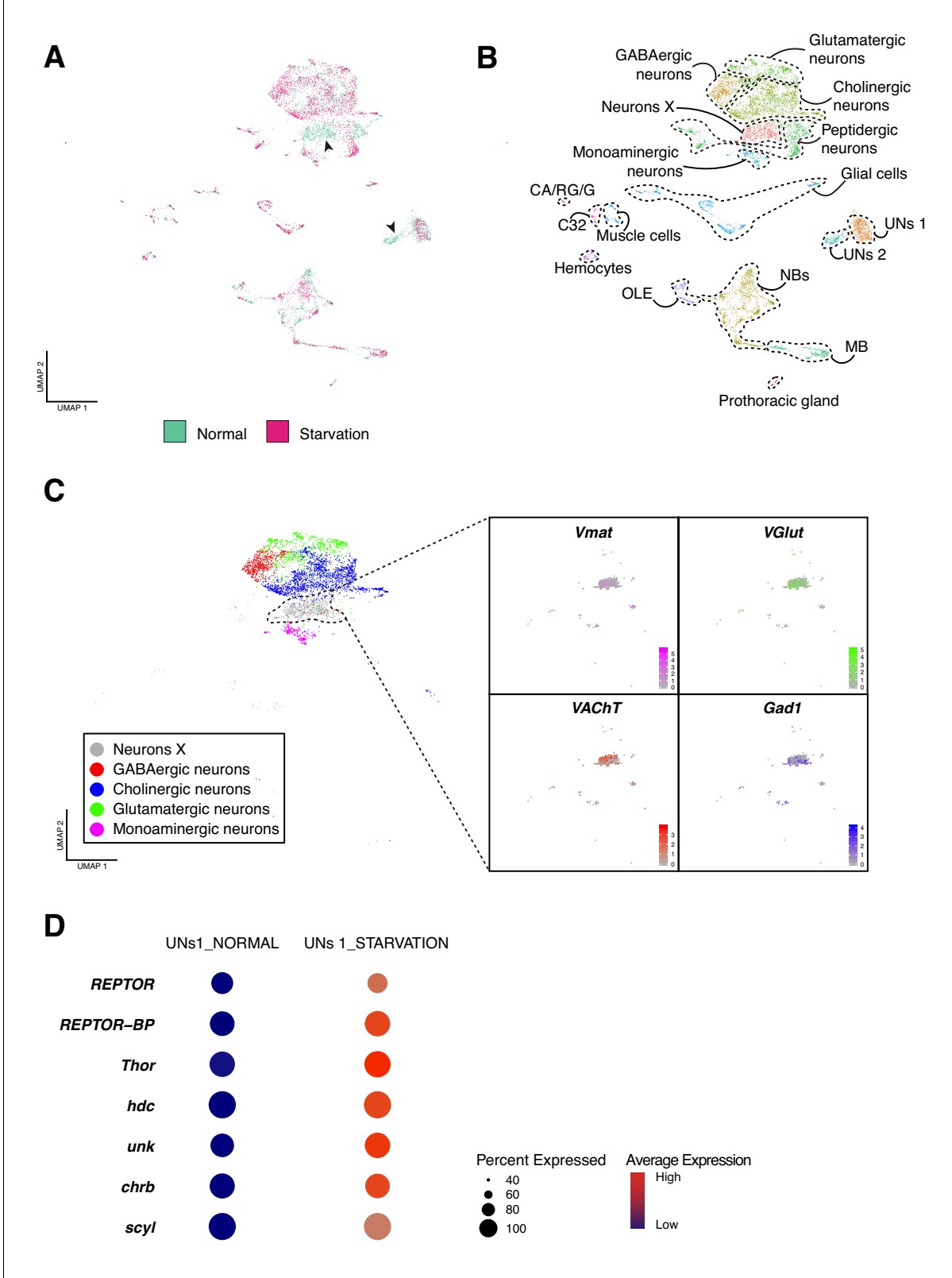

**Figure 7.** Effects of starvation on the cellular composition of the larval brain. (**A**) Starvation reveals sensitive cell types. Overlapping UMAP plots for the normal and starved condition, evidence the overall similarity among cell atlases, with few exceptions. (**B**) Integration across conditions recapitulates the principal cell types largely described in the previous sections. (**C**) A closer look to the Neurons X reveals its composition. Different neurotransmitters seemed to be equally expressed in the cells of Neurons X cluster. (**D**) UNs 1 and UNs 2 shared common features. The dot plot representation illustrates
*Figure 7 continued on next page*

*Figure 7 continued*

some of the marker genes for this particular cell type and the effect of starvation on their expression levels. Dot size corresponds to percentage of cells expressing a particular gene, while color intensity represents gene expression levels.

The online version of this article includes the following source data and figure supplement(s) for figure 7:

**Source data 1.** Clusters composition.
**Figure supplement 1.** Starvation and survival rate.
**Figure supplement 2.** Undifferentiated neurons are significantly decreased in cell number during nutrient restriction.
**Figure supplement 3.** Neuroprogenitor cells fail to exit quiescence upon starvation.

These results indicate that under nutrient-stressful conditions, glial cells potentiate catabolic pathways to survive, and pause anabolic pathways to avoid energy waste.

Next, to determine genes up or downregulated upon starvation, we scatter plotted the average expression of cells under normal feeding condition and starvation (*Supplementary file 2*, *Source data 1*). To identify outliers, we calculated the average fold change across conditions. We observed that the expression profile of differentiated cells, like neurons releasing neurotransmitters, remained invariable; indicating that their expression programs were not affected by the lack of nutrients. We found that upon food deprivation many neuropeptides were upregulated, which are known to regulate feeding behavior. *Leucokinin* (*Lk*) and *Adipokinetic hormone* (*Akh*) expression appeared upregulated after starvation, a predictable result since *Lk* negatively regulates food intake and *Akh* triggers accumulation and availability of storage lipids and glycogen (*Gáliková et al., 2015*; *Yurgel et al., 2019*). Moreover, the neuropeptides *Hug* and *SIFa*, controlling growth and metabolism and promoting feeding behavior upon hunger signals (*Martelli et al., 2017*), respectively; were both upregulated after starvation. Additionally, the upregulated neuropeptides identified in our dataset were also regulating water homeostasis and locomotor activity, as it was the case for the *Ion transport peptide* (*ITP*). In contrast, other neuropeptides were downregulated as *Allatostatin A* (*AstA*), which has been previously reported to reduce feeding and promote sleep, regulated by *Pdf* expression (*Chen et al., 2016*) (*Figure 8B*). These results suggest that cells involved in feeding behavior and homeostasis are more sensitive to starvation than the remaining cell types, and that genes associated to these mechanisms show the strongest and fastest response to food deprivation in order to alert larvae about their stressful environment.

Additionally, we observed different lncRNAs being affected by the lack of nutrients. Specially, we noticed that *lncRNA:CR40469 and lncRNA:CR42862* were upregulated in starved glial cells and NPCs (*Figure 8C*). Similarly, we detected changes in the expression levels of *lncRNA:CR42491 and lncRNA:CR44832* in NPCs and glial cells upon starvation (*Figure 8—figure supplement 1B*).

Then, we tested whether components of the TORC1 signaling pathway were affected in the larval brain upon starvation. The TORC1 complex, composed by *Target of rapamycin* (*Tor*) and *Raptor* (*raptor*), was lowly expressed in both conditions in different cell-types of the larval brain. Since TORC1, under nutrient stress condition, is inactive (*Tiebe et al., 2015*); we wondered if the change was detectable already at mRNA levels. We did not observe substantial changes at the transcriptional level upon starvation. However, as previously mentioned, two genes known to be acting downstream of TORC1: *REPTOR* and *REPTOR-BP,* showed to respond to starvation. Upon TORC1 inactivation, *REPTOR* is dephosphorylated and translocated to the nucleus, where it binds *REPTOR-BP* to form a complex and mediate transcriptional responses (*Tiebe et al., 2015*). We found that *REPTOR* expression levels were not significantly different upon starvation, but we noticed an increase in the number of cells expressing this gene, particularly in glial cells. To confirm these findings, we investigated the expression levels of the genes downstream of the *REPTOR/REPTOR-BP* complex. U*nk*, *Thor*, *nop5*, *CG16721*, *CG6770* and *CG11658* among other genes were affected upon starvation, as the number of positive cells for these genes showed to be different after starvation (*Figure 8D*). Finally, knowing that *REPTOR and forkhead box, sub-group O* (*foxo*) have overlapping target genes (*Tiebe et al., 2015*), we expected to observe an upregulation of *foxo* upon starvation. Indeed, we detected higher levels of *foxo* transcripts in this condition (*Figure 8D*). Thus, we could not detect changes in the expression levels of TORC1 components, but we did observe responses due to starvation in genes acting downstream of it, possibly indicating the presence of additional mechanisms mediating this activation.

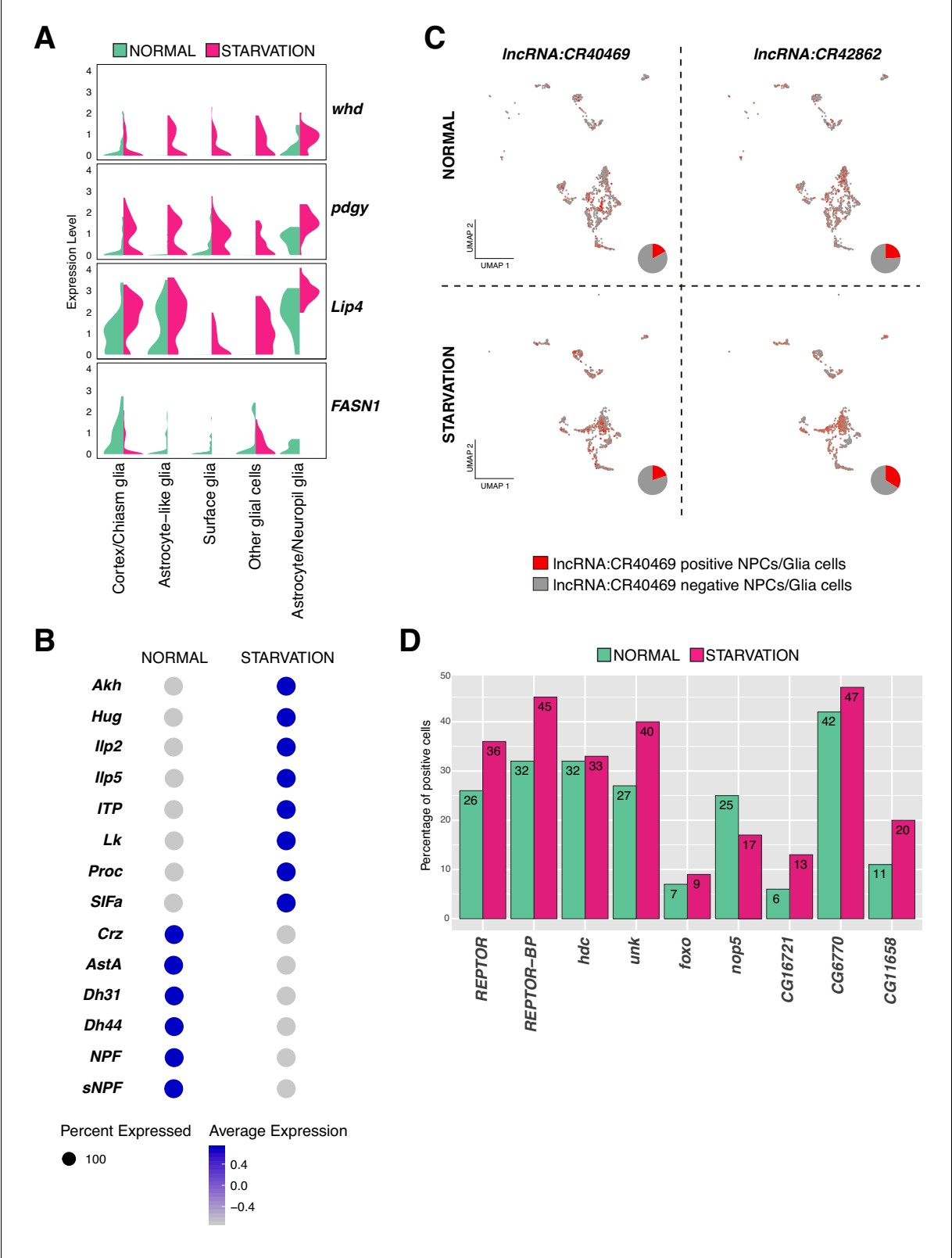

**Figure 8.** Variation in gene expression upon starvation. (**A**) Genes involved in fat acid metabolism are up and downregulated upon starvation in glial cells. Composed violin plot showing changes in gene expression across experimental conditions. Violin plots represent the probability density of the data integrated with kernel density estimation. Wider sections represent higher probability of cells with the indicated gene expression level, while skinnier sections represent lower probability. Each violin plot is split by condition, half corresponds to the normal condition and half to starved

*Figure 8 continued on next page*

*Figure 8 continued*

condition. Conditions are color coded. (B) Neuropeptides respond differently to nutrient restriction, some of them were upregulated and some others downregulated. Blue indicates high expression, while gray indicates low expression. (C) lncRNAs are also affected by starvation. lncRNA:CR40469 and lncRNA:CR42862 positive cells vary after nutrient restriction. In red, cells expressing a particular lncRNA; in gray, remaining cells. Pie graphs represent the percentage of lncRNA positive cells in the total of NPCs and glial cells. (D) Genes downstream of REPTOR/REPTOR-BP complex are affected upon starvation. Barplot showing changes in the number of cells expressing a particular gene, upon TORC1 downregulation due to starvation. Experimental conditions are color coded.

The online version of this article includes the following figure supplement(s) for figure 8:

**Figure supplement 1.** lncRNAs are sensitive to effects of starvation.

## Discussion

Here we have introduced an extensive cell atlas of the first instar larval brain at single cell resolution applying 10X Genomics technology, across two different conditions: normal feeding and starvation. Single-cell RNA sequencing has innovated the way in which tissues, organs and entire organisms are studied. Accessibility to the transcriptome of individual cells results in cell-type recognition and a better understanding of cellular processes and biological mechanisms. But large tissues display a greater number of cells, while smaller organisms are capable of performing a variety of tasks with a simplified cellular composition. Thus, *Drosophila melanogaster* has been chosen to better understand the composition of the brain. While the adult fly brain has been extensively described, we decided to fill the existent gap regarding the composition of the brain at early developmental stages. With only about 2000 cells in the brain, a more complete coverage may be achieved more rapidly. Our data set comprises about 9400 cells, suggesting that we have likely close to a 5x cell-coverage of the larval brain.

Unsupervised cluster analysis led us to identify major cell-type categories as well as more resolved cell-types. Canonical markers (e.g. *Notch*, *elav*) grouped all cells into major categories, which based on subtype makers (e.g. *ChAT*, *dpn*, *alrm*) further divide them into different cell-types. This analysis further allowed the depiction of putatively novel and intriguing biological features. An example of this is the multiple-neurotransmitter-releasing neurons, which express more than one neurotransmitter, as previously found in the adult brain (*Croset et al., 2018*). Even though these cells represent a small fraction of the total of cells populating the larval brain, it is interesting to understand the need of a neuron to produce different neurotransmitter types. Even more, deciphering the biological processes, in which these neurons are involved and the regulation behind the co-releasing phenomenon, opens a new field that requires further investigation. Similarly, co-expression of neurotransmitters and neuropeptides appears to be a common feature among peptidergic cells, suggesting different modalities of intercellular communication.

Interestingly cells of the mushroom body are closer to the neurogenic population in the UMAP plot than to the remaining differentiated neurons. Looking in detail to this group of cells, one can observe cells at different stages of differentiation: neuroprogenitors, cells undergoing differentiation and mature neurons. Knowing that the mushroom body neuroblasts continuously divide from the embryonic to the late pupal stage, and that they produce a specific neuron type (*Kunz et al., 2012*), could partially explain the localization of this cluster of cells closer to undifferentiated cells. In addition to the MBNBs, which are constantly proliferating, a subset of NPCs showed to be actively replicating its genomic content at 16 hr ALH, indicating that neuroblasts in the brain may exit their quiescent state earlier than what has been previously reported.

An intriguing, yet not fully understood, cell-type are the UNs, which in the UMAP plot locate in close proximity with NPCs. While these cells show expression of neuronal differentiation markers such as *elav* or *Syt-1*, they lack expression of genes required for neurotransmitter synthesis. Similarly, a recent study revealed a new quite abundant, neuronal cell-type. These cells identified as small undifferentiated neurons (SUs), are characterized by reduced dimensions, a dense heterochromatin and the absence of synapses (*Andrade et al., 2019*). Therefore, we speculate that our UNs may correspond to the previously described SUs. It is intriguing to speculate that *hdc* and *unk*, two genes previously described to restrict cell cycle progression in response to nutrition restriction (*Li et al., 2019*), present in neurogenic cells but enriched in UNs, have a role in maintaining these cells undifferentiated.

Moreover, we explored the changes in the larval brain caused by starvation at single-cell resolution. Food intake has a critical effect on survival and development. Nevertheless, how the brain composition and cellular programs are affected upon food deprivation remains largely unknown. The cellular content showed to be almost invariable, only few clusters disappeared after starvation. The remaining cell-types clustered together, independently of the experimental condition, indicating that cell-type identity is maintained once it has been established. Moreover, and as it was observed with aging (*Davie et al., 2018*), the effects of starvation are cell specific; transcriptional programs in neurons remained unaffected, while glial cells, NPCs and undifferentiated neurons experienced changes at the transcriptional level or even disappear due to stressful conditions. Additionally, the most susceptible genes are those involved in fat metabolism as well as some lncRNAs, which we hypothesize may have a regulatory function. Neuropeptides are also sensitive to nutritional levels. As expected, when cells face nutrient deficiency, protein synthesis is affected. In this case, cells favor the production of those neuropeptides involved in food intake and water homeostasis, downregulating the remaining ones in an attempt to survive. Nevertheless, is important to also consider and not exclude the possibility that these cell types could have been loss during sample preparation and subsequent manipulation. To confidently address this point, further repetitions are required.

Starvation is directly linked to TORC1, which integrates information about nutrient conditions of individual cells to trigger physiological responses (*Dibble and Manning, 2013*). We did not observe substantial changes at the transcriptional level for the genes in the TORC1 upon food deprivation, suggesting that cells regulate TORC1 activity at the protein level rather than at the transcriptional level. It is interesting to analyze those genes downstream of TORC1. We focused on the TORC1 signaling pathway, rather than in the insulin pathway, in an attempt to better understand nutrient sensing responses in a cell-autonomous fashion and not at a systemic level. Recently, REPTOR and REPTOR-BP were shown to be transcription factors responsible for mediating some of the responses of reduced TORC1 activity (*Tiebe et al., 2015*). As expected, REPTOR and its binding protein are both upregulated upon starvation, as well as their target genes. Under limited nutrient conditions, larvae have to choose between anabolic or catabolic processes. Here, we observe that under our extreme starvation condition, larvae have no other option than to catabolize their storages to obtain some energy and survive as long as possible. Even though intracellular nutrient-sensing activates TORC1, extracellular sensing must also be required to mount a complete physiological response. In this way, integration between different signaling pathways, Insulin-Foxo and TORC1-REPTOR, must be required.

Furthermore, the availability of the wildtype brain cell atlas as well as the starvation induced changes provide a resource for further functional analysis of neural circuits of the larval brain as well as developmental studies at the fundamental unit of biological organization: the cell (*Regev et al., 2017*).

## Materials and methods

**Key resources table**

| Reagent type (species) or resource | Designation | Source or reference | Identifiers | Additional information |
|---|---|---|---|---|
| Antibody | Anti-GFP (Chicken polyclonal) | Abcam | Cat# ab13970, RRID:AB_300798 | IF(1:1000) |
| Antibody | Anti-GFP (Rabbit polyclonal) | Thermo Fischer | Cat# A-6455, RRID:AB_221570 | IF(1:1000) |
| Antibody | Anti-Dpn (Guinea pig polyclonal) | *Spéder and Brand, 2014* | | IF(1:5000) |
| Antibody | Anti-Dlg (Mouse monoclonal) | Iowa H.B. | Cat# 4F3, RRID:AB_528203 | IF(1:100) |
| Antibody | Anti-Fas2 (Mouse monoclonal) | Iowa H.B. | Cat# 1D4, RRID:AB_528235 | IF(1:20) |
| Antibody | Anti-hdc (Mouse monoclonal) | Iowa H.B. | Cat# U33, RRID:AB_10659722 | IF(1:5) |

*Continued on next page*

*Continued*

| Reagent type (species) or resource | Designation | Source or reference | Identifiers | Additional information |
|---|---|---|---|---|
| Antibody | Anti-repo (Mouse monoclonal) | Iowa H.B. | Cat# 8D12, RRID:AB_528448 | IF(1:20) |
| Antibody | Anti-prt (Rabbit polyclonal) | David E. Krantz | | IF(1:300) |
| Antibody | Anti-Tyrosine Hydroxylase (Rabbit polyclonal) | Millipore | Cat# AB152, RRID:AB_390204 | IF(1:100) |
| Antibody | Anti-Elav (Rat monoclonal) | Iowa H.B. | Cat# 7E8A10 | IF(3:100) |
| Antibody | Anti-DsRed (Rabbit polyclonal) | Takara Bio | Cat# 632496, RRID:AB_10013483 | IF(1:1000) |
| Antibody | Anti-Rabbit Alexa 647 (Goat polyclonal) | Molecular Probes | Cat# A-21244, RRID: AB_2535812 | IF(1:200) |
| Antibody | Anti-Rabbit Alexa 488 (Goat polyclonal) | Molecular Probes | Cat# A-11008, RRID: AB_143165 | IF(1:200) |
| Antibody | Anti-Mouse Alexa 647 (Goat polyclonal) | Molecular Probes | Cat# A-21235, RRID: AB_2535804 | IF(1:200) |
| Antibody | Anti-Guinea pig Alexa 488 (Goat polyclonal) | Molecular Probes | Cat# A-11073, RRID: AB_2534117 | IF(1:200) |
| Antibody | Anti-Chicken Alexa 488 (Goat polyclonal) | Molecular Probes | Cat# A-11039, RRID: AB_2534096 | IF(1:200) |
| Antibody | Anti-Rat Alexa 647 (Goat polyclonal) | Molecular Probes | Cat# A-21247, RRID: AB_141778 | IF(1:200) |
| Antibody | Anti-Guinea pig Alexa 647 (Goat polyclonal) | Molecular Probes | Cat# A-21450, RRID: AB_2535867 | IF(1:200) |
| Genetic reagent (*D. melanogaster*) | VGlut-Gal4 | BDSC | RRID: BDSC_24635 | |
| Genetic reagent (*D. melanogaster*) | UAS-myrGFP | BDSC | RRID: BDSC_32198 | |
| Genetic reagent (*D. melanogaster*) | Gad1-GFP | BDSC | RRID: BDSC_59304 | |
| Genetic reagent (*D. melanogaster*) | VAChT-Gal4 | BDSC | RRID: BDSC_39078 | |
| Genetic reagent (*D. melanogaster*) | UAS-mCD8::RFP | BDSC | RRID: BDSC_32219 | |

## Fly strains

*Drosophila melanogaster* Canton-S was used as the wild type strain. Other fly strains used are described in the key resources table.. All flies were kept at 25°C in a 12 hr/12 hr light-dark cycle in vials containing conventional cornmeal agar medium.

## Larval culture

Larvae were grown and kept at 25°C in the same conditions as the adult flies and late first instar larvae were collected at 16 hr after larval hatching (ALH). For the starved condition, larvae were collected at 12 ALH, quickly washed to remove food leftovers and transferred to a Petri dish with 2% of agar, humidified with PBS, for 4 hr at 25°C.

## Survival rate

Larvae were grown as described before and at 12 ALH transferred to a Petri dish with 2% of agar, humidified with PBS at 25°C. After 4 hr of starvation, dead and alive larvae were counted and the percentages were calculated based on the initial number of larvae in the Petri dishes.

## Immunofluorescence

First instar larval brains were dissected in ice-cold PBS, placed on a 22 × 22 cover slip and fixed in 4% paraformaldehyde for 18 min at room temperature with slow agitation. Fixed brains were washed three times in PBST (PBS 0,3% Triton X-100), for 20 min each, at room temperature with slow agitation. All these steps were performed in a Columbia staining jar. Primary antibodies (see Key resources table) were prepared in PBST and the cover slips with the brains were incubated overnight at 4°C in a humid chamber. Primary antibody solutions were removed and brains were washed again three times in PBST for 20 min each at room temperature with slow agitation. Next, the secondary antibody (see Key resources table) solutions were added and incubated as described. After overnight incubation, secondary antibody solutions were removed and washes with PBST were performed. Brains were mounted in Vectashield (Vector Laboratories) antifade mounting medium. Images were acquired using Leica TCS SP5 confocal microscope and images were assembled using Fiji and Adobe Illustrator CC 2018. DAPI was added together with the secondary antibodies.

## EdU incorporation

Larval brains were dissected in ice-cold PBS and immediately incubated in PBS containing 200 µg/ml of EdU (Thermo Fischer Scientific - C10637) for 30 min at room temperature. Brains were quickly washed with PBS and fixed in 4% paraformaldehyde for 18 min at room temperature. Upon fixation, brains were washed with PBST and EdU detection was performed following the manufacturer's recommendations. *Dpn* antibody staining was executed as described above. The proliferation index (PI) was calculated with following formula.

$$PI = \frac{Dpn^+ EdU^+}{Dpn^+}$$

## Brain dissection, dissociation and single cell suspension

First instar larvae (40-45) were collected and quickly washed in water to remove food leftovers and yeast. Larvae were placed in drops of ice-cold PBS on the inside of a plastic petri dish lid. Fine forceps were used to dissect the larval brain. Once the larval brain was exposed, the ventral nerve cord was cut out using a pair of pin holders (*Figure 1A*), and the intact brain-lobes were collected in a low DNA binding tube containing 250 µl of ice-cold RNA free PBS for a maximum of one hour. The tubes containing the brains were centrifuged at 2000 rpm for 5 min at 4°C. After centrifugation, the supernatant was carefully removed and replaced with 200 µl of *collagenase* (1 mg/ml Sigma-Aldrich C9722) and incubated for 1 hr at 25°C with continuous agitation. To guarantee full brain digestion, the suspension was pipetted up and down each 10 min. The enzymatic reaction was arrested by diluting the suspension with 1 ml of PBS 0.04% BSA (Thermo Fischer Scientific AM2616). After washing the cells, the cell suspension was filtered through a 40 µm Flowmi Cell strainer (Bel-Art H13680-0040). The filtered suspension was centrifuged at 2000 rpm for 5 min at 4°C. The supernatant was discarded, cells were resuspended in 50 µl of PBS 0.04% BSA and further dissociation was ensured by gently pipetting the entire volume, approximately 200 times. Cell concentration was determined using a hemocytometer (Neubauer improved – Optik Labor) under a Leica DM 100 led microscope.

## 10x genomics and sequencing

scRNA-seq libraries were prepared using the Chromium Single Cell 3' Library and Gel Bead Kit v3 (10X Genomics), according to the manufacturer's protocol (User Guide). Chips were loaded after calculating the accurate volumes using the 'Cell Suspension Volume Calculator Table'. With an initial single-cell suspension concentration equal to 1000 cells/µl, we targeted to recover approximately 10,000 cells. Once GEMs were obtained, reverse transcription and cDNA amplification steps were performed. Sequencing was done on Illumina NovaSeq 6000 S2 flow cell generating paired-end reads. Different sequencing cycles were performed for the different reads, R1 and R2. R1, contained 10X barcodes and UMIs, in addition to an Illumina i7 index. While R2 contained the transcript-specific sequences.

## 10x data processing

The sequenced libraries were processed according to Cell Ranger (version 2.2.0) count and aggr (aggregation) pipelines, provided by 10X Genomics. The reference genome was built based on the

3$^{rd}$ 2018 FlyBase release (*D. melanogaster* r6.22). The number of cells detected at the end of each experiment was determined by Cell Ranger with the number of barcodes associated with cell-containing partitions, estimated from the barcode UMI count distribution. Two final datasets were obtaining upon aggregation of individual experiments, non-starved and starved datasets, producing single feature-barcode matrices. The first one was built aggregating the libraries from three biological replicates, corresponding to larvae kept in conventional cornmeal agar medium prior to dissection (considered as the normal condition). This dataset resulted in a total of 4708 cells with a median of 1434 genes per cell. The second dataset was built in a similar way, but aggregating libraries from two biological replicates, corresponding to larvae starved for 4 hr prior to dissection (considered as the starvation condition), resulting in a dataset of 4645 cells with a median of 1962 genes per cell. The different aggregations were carried without specifying any normalization mode.

## Seurat data processing

Seurat version 3.0 (*Butler et al., 2018*; *Satija et al., 2015*; *Stuart et al., 2018*) pipeline was adapted and executed on the normal (non-starved) dataset and on a combination of both conditions: 'normal' and 'starvation' datasets. The matrices produced by cell ranger were processed and duplets, or eventually multiplets, were discarded based on the overall gene expression per cell. Cell quality was assessed by the percentage of mitochondrial gene expression per cell. Thus, cells with unique feature counts between 200 and 4500, and with less than 20% of mitochondrial genes were kept for downstream processing. Additionally, genes expressed in at least one cell were considered for the analysis. The final processed datasets resulted in 4349 and 4347 cells with a total of 12,942 and 13,589 identified genes, for the non-starved and starved datasets, respectively. Once Seurat objects were built, pre-processing steps were performed before downstream analysis. First, a log-normalization with a scale factor of 10,000 was applied to normalize gene expression of individual cells by the total gene expression of each dataset. Second, a linear transformation was executed to remove unwanted source of variation. Lastly, highly variable genes were determined applying FindVariableFeatures function with default parameters, producing a total of 2000 variables genes following the R package developer's recommendations.

Upon preprocessing, the highly variable genes were considered for the dimensional reduction analysis to highly biological significance. To define the true dimensionality of the dataset, several approaches were considered: Elbow-Plot and JackStraw-Plot tests together with an evaluation of PC-heatmaps. Finally, principal components (PCs) were selected visually by carefully inspecting Elbow-Plots (*Supplementary file 3*) in order to assess the percentage of variance explained by each PC. In this way, 31 PCs were considered to identify cell clusters with a graph-based approach. To better resolve clusters, we modified the resolution, as increasing this parameter helps to subdivide existing clusters and gain granularity. Therefore, we used a resolution equal to two based on the number of cells and a carefully inspection of the dataset. Then, a non-linear dimensional reduction was performed to visualize the results in UMAP plots. Lastly, cluster identities were assigned after determining top 10 differentially expressed genes across cell clusters using a Wilcoxon Rank Sum test. Some of these clusters were later subset and re-analyzed for further characterization.

Integration between conditions was performed by identifying common anchors across both datasets, normal and starvation, to later combine them into a single Seurat object by applying the 'IntegrateData' function, which produced a batch-corrected expression matrix. The pipeline was applied with default parameters. Downstream analysis was performed as described above. Finally, Identification of differentially expressed genes across conditions was achieved by comparing the expression profile of each cluster across experimental conditions. These results were later represented in scatter plots to visualize outliers. For simplicity, only genes with a fold change higher than one or smaller than −1 were labeled in these representations. In the case where many genes resulted as outliers, only the top 10 genes with the highest scored were included in the graphs (*Supplementary file 2*, *Source data 1*).

## Cell-cycle scoring

Cell cycle phase scores were assigned by applying 'CellCycleScoring' function from Seurat R package, by providing a list of marker genes for S and G2 and M (G2/M) phases. Cells that did not express marker genes for neither phases S nor G2/M, were considered to be in G1 phase.

## Re-clustering and further seurat analyisis

### Neural progenitor cells

Cells characterized as neurogenic were reanalyzed and subclustered, keeping only those cells with more than 200 expressed genes per cell. The data was normalized and a new PCA was computed. Then, 13 PCs were chosen, as previously described, and clusters were visualized in a UMAP plot, with a resolution equal to 1. Identities were assigned after analyzing differentially expressed genes, following the same principle described above.

### Glial cells

For glial cells, a similar analysis was performed. In this case, 11 PCs with a resolution equal to one were selected to generate a UMAP plot and represent new clusters. Cell identities were assigned after visually evaluating enriched marker genes expression.

### Mushroom body

MB cells were reanalyzed, subclustered and identified, as it was described for NPCs. This time 10 PCs were chosen and clusters were visualized in a UMAP plot, with a resolution of 0.8.

### Peptidergic neurons

Peptidergic neurons were reanalyzed individually. This particular analysis only considered cells with high expression levels of the peptide of interest. In this way, IPCs and PDF, PTTH and CRZ neurons were selected as follows: Ilp2 >4, Pdf >4, PTTH >3 and Crz > 6, respectively. Subsequent analysis was performed based on the expression of reported marker genes for each cell-type.

## Acknowledgements

We are grateful to Boris Egger, Cornelia Fritsch, Lucia de Andres Bragado, Jules Duruz and other members of the Sprecher lab for discussion and sharing advices. We thank the Next Generation Sequencing (NGS) Platform at the University of Bern and the Light Microscopy and Image Analysis Facility at the University of Fribourg for providing the necessary equipment and technical support for our experiments, as well as the Developmental Studies Hybridoma Bank at the University of Iowa and the Bloomington Stock Center for sharing reagents and flies. We would also like to thank David E Krantz for providing anti-*Prt* antibody. This work was supported by the Swiss National Science Foundation.

## Additional information

### Funding

| Funder | Grant reference number | Author |
| --- | --- | --- |
| Schweizerischer Nationalfonds zur Förderung der Wissenschaftlichen Forschung | 31003A_149499 | Simon G Sprecher |
| Schweizerischer Nationalfonds zur Förderung der Wissenschaftlichen Forschung | SystemsX- SynaptiX RTD | Rémy Bruggmann Simon G Sprecher |
| Schweizerischer Nationalfonds zur Förderung der Wissenschaftlichen Forschung | 310030_188471 | Simon G Sprecher |

The funders had no role in study design, data collection and interpretation, or the decision to submit the work for publication.

### Author contributions

Clarisse Brunet Avalos, Formal analysis, Investigation, Visualization, Methodology, Writing—original draft; G Larisa Maier, Methodology; Rémy Bruggmann, Conceptualization, Data curation, Software,

Funding acquisition, Methodology; Simon G Sprecher, Conceptualization, Supervision, Funding acquisition, Project administration, Writing—review and editing

### Author ORCIDs
Clarisse Brunet Avalos ![ORCID] https://orcid.org/0000-0002-7781-004X
Rémy Bruggmann ![ORCID] https://orcid.org/0000-0003-4733-7922
Simon G Sprecher ![ORCID] https://orcid.org/0000-0001-9060-3750

### Decision letter and Author response
Decision letter https://doi.org/10.7554/eLife.50354.sa1
Author response https://doi.org/10.7554/eLife.50354.sa2

## Additional files

### Supplementary files
• Source data 1. List of differentially expressed genes across different feeding conditions.Table displaying the average log-fold change values for the list of differentially expressed genes among clusters and conditions.

• Supplementary file 1. Sequencing metrics.Table displaying sequencing details for each of the biological replicates and aggregated datasets.

• Supplementary file 2. Differentially expressed genes across different feeding conditions.Scatter plots illustrating the differentially expressed genes per cluster and per condition. Dark blue: a tendency line. Light-dashed line: FC=±1.

• Supplementary file 3. Data dimensionality.Elbow plots analyzed to select the real dimensionality of the datasets. In red and pointed with an arrow, the number of PCs selected for downstream processing.

• Transparent reporting form

### Data availability
The single-cell sequencing data has been deposited in GEO under the accession code GSE134722.

The following dataset was generated:

| Author(s) | Year | Dataset title | Dataset URL | Database and Identifier |
|---|---|---|---|---|
| Clarisse Brunet Avalos, G Larisa Maier, Rémy Bruggmann, Simon G Sprecher | 2019 | Single cell transcriptome atlas of the Drosophila larval brain | https://www.ncbi.nlm.nih.gov/geo/query/acc.cgi?acc=GSE134722 | NCBI Gene Expression Omnibus, GSE134722 |

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
