## [Decision Letter]

**Acceptance summary:**

In this study, the authors lay out single-cell transcriptome maps in the first-instar larval brain of the fruitfly, *Drosophila*. This extensive and substantial map covers two central nervous system lobes. Single-cell RNAseq in this study illuminates a more representative transcriptional landscape of cellular composition than other similar studies. By analyzing sequence data, the authors also confirm that the maps are consistent with information available in the literature, highlighting that the resource offered by the study will be a useful and dependable reference for future studies. This work will be a considerable contribution to the field of neurogenesis and 'metabolic-neuroscience'.

**Decision letter after peer review:**

Thank you for submitting your article "Single-cell transcriptome atlas of the *Drosophila* larval brain" for consideration by *eLife*. Your article has been reviewed by three peer reviewers, and the evaluation has been overseen by a Reviewing Editor and K VijayRaghavan as the Senior Editor. The reviewers have opted to remain anonymous.

The reviewers have discussed the reviews with one another and the Reviewing Editor has drafted this decision to help you prepare a revised submission.

Summary:

The authors lay out single-cell transcriptome maps in the first instar larval brain under two conditions – nutritionally normal and starved. This attempt is very extensive and substantial, as the map covers ~1.75x of two CNS lobes. Single-cell RNAseq in this study illuminates more representative transcriptional landscape of cellular composition than other similar studies (for example, Davie et al., 2018 in Cell- more sequenced cell numbers but 1x coverage) by exploiting the model organism with reasonable neuronal complexity and the size benefit. By analyzing the seq data, Avalos et al. confirmed that the maps are consistent with information available in the literature, highlighting that the resource offered by the study will be a useful and dependable reference for future studies.

In the cells from the nutritionally normal condition the authors identify 14 clusters of cells that correspond to five cell types – 1) Neural stem cells, 2) Differentiated neurons, 3) Immature neurons 4) Glia, and 5) Non-neural cells.

Amongst the neural stem cells, the author distinguished optic lobe precursors and all other stem cells. They focus on neurotransmitter identity of the neurons and show that glutamatergic neurons are the most abundant, followed by cholinergic, GABAergic and monoaminergic. They find low levels of co-expression of neurotransmitters, which has also been reported in other scRNA data from adult brains. They also identify the peptidergic PDF neurons, insulin-producing cells, and corazonin positive cells and identity some markers for some of them. They show that most of the peptidergic neurons are also glutamatergic. In their glial cluster, they identify surface glia, cortex glia, astrocytes and chiasm glia based on known molecular markers.

Finally, they analyse the cells from the starved condition. Interestingly, they identify two clusters that were absent upon starvation. One of these clusters consists of differentiated neurons of various neurotransmitter identities, the other of immature neurons – they don't discuss these further. Instead, they look at the transcriptome of all remaining cell types and show that genes involved in fatty acid metabolism are upregulated and those involved in fatty acid synthesis are downregulated in glial cells. They also show upregulation, downregulation and 'mis'-regulation of many neuropeptides and long non-coding RNAs in many cells, including glia upon starvation.

While the paper can be a considerable contribution to the field of neurogenesis and metabolic neuroscience, several concerns need to be addressed. In particular, the description of the experiments are inadequate and this makes it hard to properly assess the quality of the data, and therefore the results.

Essential revisions:

1) Because the entire manuscript relies on single-cell sequencing, we would like to have seen a more rigorous description of the quality of data they obtained. It's possible that the authors have done these quality checks but have not mentioned it in the manuscript. But, given that this is a rapidly improving technology, we would request the authors to please present this aspect of their data. For example:

– Have they done replicates of their experiments? How were batch effects between the two runs (which could arise due to various factors such as changes in sequencing depth, RNA capture efficiency, RNA->cDNA conversion efficiency, etc.) accounted for, particularly when comparing gene expression between normal and starved conditions? Was this automatically done by Seurat, and if so, was this done with Seurat's default settings?

– How many reads do they have per cell? Based on the text, the numbers don't add up. If the authors have 764,000 reads in total, they have anywhere between 70-200 reads per cell (depending on the answer to the next question). This cannot correspond to 1500 genes, unless they are pooling reads within clusters – are they?

– Have they sequenced the normal and starved brains in the same run or different runs? If the latter, then, could the authors provide more information about each individual sequencing run, namely, the median number of genes obtained in the starved and fed cases? Currently, paragraph one of subsection “Single-Cell RNA-Seq of the *Drosophila* larval brain reveals different categories of characteristic cell-types” does not convey information about the individual libraries.

– 3000 odd cells per condition seem rather low given that other similar experiments get 20,000-60,000 (Corset et al., 2018; Davie et al., 2018; Konstntinides et al., 2018).

2) The authors identify 28 clusters of cells, which they manually annotate. Their source data shows the top 10 genes they've used for this purpose. Could the authors please edit this table to include (a) the names of each cluster, (b) whether the markers used are based on prior literature or reported in this work based on in-silico considerations, and (c) the number of cells in each cluster for both the conditions?

3) The Supplementary file 2 shows average gene expression as scatter plots for each cluster of cells of the normal and starved conditions. These plots are very useful in visually identifying outliers. However, the criterion for identifying outliers seems to change between panels. Could the authors please explain what made them pick some genes on the outer boundaries of the scatter-plot as outliers, and others not? To clarify this, could the authors please compute the fold-change in mean expression between fed- and starved- (or the other way around) conditions and add this information to the statements made in paragraph two of subsection “Transcriptional responses to starvation” –. Alternately, they could define a criterion to define outliers as genes where the fold-change in mean expression between fed- and starved- conditions was greater than X, where X is 2 or 3?

4) The loss of the UN-2 and neurons^-1^ clusters is quite striking! An expected response to starvation at that stage is the failure of neural stem cells to exit quiescence and re-enter cell division. This could show up as clusters of cells missing. However, more mundane possibilities that would be useful to rule out would be: (1) that UN-2 and neurons^-1^ are cell types that are very rare in the larval brain, and hence, could stochastically be lost during sample processing from the starved larvae, and (2) that cells from the UN-2 and neurons^-1^ clusters are indeed present in the data, but due to a low read depth from those cells, have been assigned to other clusters that are present in the starvation condition?

5) In the VNC, the neural stem cells that exit quiescence at 15hours ALH are the G2 arrested stem cells. Have the authors considered looking at their neural stem cell cluster in the fed vs starved conditions to see if they can distinguish G0 and G2 arrested stem cells, and the effect of starvation on them?

6) Are there any molecular markers in the UN-2 and neurons^-1^ clusters, that the authors could use in vivo to demonstrate the loss of those cell types upon starvation?

7) A few aspects in the description of the methodology is lacking in detail in order to be able to reproduce the analyses in the paper for future readers. Specifically:

– Results section “Altogether, our datasets displayed a median of 764,000 reads and

1500 genes per cell, and in total of about 12,000 of genes were detected”; – Is this the total number of reads, or the number of mapped reads?

– Paragraph three of subsection “Single-Cell RNA-Seq of the *Drosophila* larval brain reveals different categories of characteristic cell-types” –: Can the authors state how many cells (what proportion) finally fell into non-neuronal clusters?

– Paragraph two of subsection “Mapping neural progenitor cell diversity reveals presence of characteristic neurogenic cell-types” –: How was the similarity of expression profiles quantified? Was it in terms of which genes had non-zero expression? "…examined the top differentially expressed genes…"? Is this the top 10 differentially expressed genes, or a different number than 10?

– Subsection “Seurat data processing” –: Can the authors describe how many genes and cells were finally considered for analyses in the paper after the filtering criteria mentioned were imposed? Or are the numbers described in the first paragraph – of the Results section obtained after the filtering criteria are imposed?

– Subsection “Seurat data processing” –: The sequence of operations carried out is unclear. Was the PCA calculated by taking all genes in each cell into account, or only based on the 2000 most highly variable genes? If it was the latter, could the authors explain why the PCA wasn't done with all genes wasn't performed, and add this to the Materials and methods section? On the other hand, if the PCA was calculated with the expression of all genes in each cell, could they explain why the set of the 2000 most variable genes was constructed? Was this set of variable genes used for identifying marker genes or for clustering?

– Subsection “Seurat data processing”: – What is the linear transformation that was applied before PCA was carried out? Why was this linear transformation needed?

– In the same section – it appears that three different methodologies were considered for choosing the number of principal components. Could the authors mention which methodology was finally followed for choosing the number of principal components? It will be useful to add these plots as a supplemental figure. Alongside this, could the authors also explain the amount of variance captured each time a given number of principal components is chosen?

– UMAP resolution – could the authors add a couple of sentences explaining the impact of increasing or decreasing the UMAP resolution used for clustering? Is this supposed to aid in sub-clustering?

– Finally, in the interest of reproducibility, it would be nice to make available (through a URL) the scripts that were used to carry out the analyses in the manuscript.

8) Cell numbers: If we go with their estimate of 2000 cells per brain, and they analyse 3500 cells (after filtering), then they are sampling the brain with less than a two-fold coverage. Given the tremendous diversity in the *Drosophila* CNS it would seem that this minimal coverage risks missing a whole range of rare cell types. And this of course has major bearing on the comparison to the starvation conditions as well, which was also less than two-fold. Granted, they do identify some rare cells, such as PDF neurons, but nevertheless, a less than two-fold coverage of an extremely diverse system risks resulting in misinterpretations, especially when comparing between conditions.

9) 4-hour starvation seems very short for getting any major transcriptomic changes, and indeed the changes appear to be minor. Is there any evidence for any physiological effect of a 4-hour starvation? Why was a 4-hour starvation chosen? Previously published larval nutrient restriction schemes have been e.g., 24-36 hours (e.g., PMID 21346761). As it now stands, it is difficult to relate the transcriptional changes to any physiological effect(s), and this brings down the impact of the study. Any progress in this area would have greatly strengthened the study.

10) Related to this issue, studies by the Alex Gould and Andrea Brand labs have revolved around the role of starvation in gating how/when neural progenitors emergence from quiescence during larval stages. This has revealed input from InR, PI3K, Akt, Alk/Jeb and TOR signalling, and some unknown signals, in part provided from the periphery, possibly relayed via CNS surface glia and onto progenitors. The authors of the current study, intriguingly, do find TOR signalling components altered, but do not relate this to the previous studies from Brand and Gould. In fact, somewhat disconcertingly, none of these seminal publications, in Nature, Cell, CellRep, are even mentioned in the current study (PMIDs: 21346761, 21816278, 23478023, 21183078). By analysing responses to starvation separately in glia versus progenitors, the authors have a golden opportunity to shed further light on this intriguing issue, but do not even appear to be aware of the literature in this field.

[Editors' note: further revisions were requested prior to acceptance, as described below.]

Thank you for resubmitting your work entitled "Single cell transcriptome atlas of the *Drosophila* larval brain" for further consideration by *eLife*. The manuscript has been improved but there are some remaining issues that need to be addressed before acceptance, as outlined below:

1) In the revised version of this manuscript, the authors have addressed and satisfied most of the reviewers' concerns. They have clarified and explained most of the text that made interpretation of their data difficult. They have also added the second replicate of their starved condition. In light of these changes, their work is both interesting and valuable to the field and is recommended for publication.

A couple of important points comment, though. There might be an error in their supplementary xls sheet for marker genes. The log-fold changes and p-values associated with some genes in the xls sheet do not seem plausible: there are astronomically high log-fold changes and p-values that are greater than one. For example, consider genes #6,#8, #9 from the "cholinergic neurons 1" cluster. These genes have log-fold-changes of 129,541,248,584,619, 0.616 and 107,496,381,242,778, with associated adjusted p-values of 8.63e-35, 3.49e-17 and 6.24e07.

It's possible that these large values occur partly due to the transformations applied to the read counts of the genes in each cell.

P-values – adjusted or otherwise – cannot exceed 1, but the p-value of gene #9 is 62.4 million.

One might expect p-values to decrease as log-fold-changes increase. So, gene #8, with a fold-change of 0.616, would be expected to have a larger (less significant) p-value than gene #9, which has a fold-change of 107,496,381,242,778. But instead, we see that gene #6 has a p-value of 3.49e-17 while gene #9 has an implausible p-value of 62.4 million.

2) Introduction section: "nutriments" should be "nutrients"

3) Results section paragraph one: I find the mean reads/cell to be very high. From our own experience, and from numerous papers, you typically detect 25,000-50,000 gene reads/cell, resulting in the detection of 1000-2500 genes/cell. They list "836,393 mean reads per cell".

---

## [Author Response]

Essential revisions:1) Because the entire manuscript relies on single-cell sequencing, we would like to have seen a more rigorous description of the quality of data they obtained. It's possible that the authors have done these quality checks but have not mentioned it in the manuscript. But, given that this is a rapidly improving technology, we would request the authors to please present this aspect of their data. For example:

This is indeed an important point and we apologize for previously not including more information. We have of course performed quality checks and now included this information in the manuscript (see points below).

– Have they done replicates of their experiments? How were batch effects between the two runs (which could arise due to various factors such as changes in sequencing depth, RNA capture efficiency, RNA->cDNA conversion efficiency, etc.) accounted for, particularly when comparing gene expression between normal and starved conditions? Was this automatically done by Seurat, and if so, was this done with Seurat's default settings?

We have performed the experiments in three biological replicates for normal conditions and two biological replicates for starvation conditions. Replicates were processed with 10X Genomics’ software for single-cell RNA-seq experiments. First, each replicate was processed with “cell ranger count”. Then, all the experiments from the same experimental condition were aggregated with cell “ranger aggr”, to produce a single feature-barcode matrix containing all the data. Since we only used Single Cell Gene Expression v2, there was no need to correct for possible batch effects due to the chemistry version. Finally, the datasets were quality checked, scaled and normalized with Seurat R package.

For the comparison between conditions, we followed the recommendations given by Seurat v3 developers. We performed first an anchors identification step, to later combine the two datasets applying “IntegrateData” function, which produced a batch-corrected expression matrix. We now describe these steps in-depth in the Materials and methods section.

– How many reads do they have per cell? Based on the text, the numbers don't add up. If the authors have 764,000 reads in total, they have anywhere between 70-200 reads per cell (depending on the answer to the next question). This cannot correspond to 1500 genes, unless they are pooling reads within clusters – are they?

We apologize for not describing this clearly. In line with the previous point, we have now included this information in the Materials and methods section. We have also now included a supplement table describing the sequencing metrics for each experiment (Supplementary file 1). For the normal condition, the total number of reads is equal to 3,937,742,005 and the estimated number of cells is 4708 with 836,393 mean reads per cell.

– Have they sequenced the normal and starved brains in the same run or different runs? If the latter, then, could the authors provide more information about each individual sequencing run, namely, the median number of genes obtained in the starved and fed cases? Currently, paragraph one of subsection “Single-Cell RNA-Seq of the Drosophila larval brain reveals different categories of characteristic cell-types” does not convey information about the individual libraries.

This is indeed a valid point and we apologize for not including this information before. We have now included a supplementary table with all the sequencing metrics for each experiment (Supplementary file 1). In this file, we describe: total number of reads, mean reads per cells, expected number of cells, total number of genes detected, sequencing saturation among other parameters.

– 3000 odd cells per condition seem rather low given that other similar experiments get 20,000-60,000 (Corset et al., 2018; Davie et al., 2018; Konstntinides et al., 2018).

We have now performed additional single-cell experiments for each condition. Summarizing, now the normal and starved conditions consist of three and two biological replicas, respectively. In total, our dataset now comprises of 9353 cells. We hereby have an approximate 5x coverage of the first instar larval brain.

2) The authors identify 28 clusters of cells, which they manually annotate. Their source data shows the top 10 genes they've used for this purpose. Could the authors please edit this table to include (a) the names of each cluster, (b) whether the markers used are based on prior literature or reported in this work based on in-silico considerations, and (c) the number of cells in each cluster for both the conditions?

This is an excellent suggestion. We feel that such data may provide a valuable resource for various topics and have now included further details to Figure 1—source data 1, in order to clarify the annotation process. In this table, one can find the top 10 marker genes used to identify cell types. In a second sheet, we have specified the number of cells per cluster. And, in a third sheet, we added a table illustrating the literature-based approach used during the annotation step. In addition, we now include a new source file for Figure 7 (Figure 7—source data 1), containing the number of cells per cluster and per condition.

Moreover, to further gain experimental support for the cluster analysis and annotation we have extended the analysis of neurotransmitter co-expression. We now show immunostainings supporting co-expression of Glutamate and GABA as well as GABA and Acetylcholine (Figure 3—figure supplement 1).

3) The Supplementary file 2 shows average gene expression as scatter plots for each cluster of cells of the normal and starved conditions. These plots are very useful in visually identifying outliers. However, the criterion for identifying outliers seems to change between panels. Could the authors please explain what made them pick some genes on the outer boundaries of the scatter-plot as outliers, and others not? To clarify this, could the authors please compute the fold-change in mean expression between fed- and starved- (or the other way around) conditions and add this information to the statements made in paragraph two of subsection “Transcriptional responses to starvation”. Alternately, they could define a criterion to define outliers as genes where the fold-change in mean expression between fed- and starved- conditions was greater than X, where X is 2 or 3?

This is indeed a valid point and we apologized for the missing information. To address this point, we have now adapted the result section and included a more detailed description in the Materials and methods section. Previously, we had labeled those genes that appeared to be visual outliers. Now, we calculated the logFC and represented those with a higher value than 1 or lower than -1. In the case of a large number of outliers, we represented the top 10 genes with the highest variability. These genes are labeled in the scatter plots. Moreover, we have included a complete table of the genes (Supplementary file 2—source data 1) that are differentially regulated for all clusters.

4) The loss of the UN-2 and neurons^-1^ clusters is quite striking! An expected response to starvation at that stage is the failure of neural stem cells to exit quiescence and re-enter cell division. This could show up as clusters of cells missing. However, more mundane possibilities that would be useful to rule out would be: (1) that UN-2 and neurons^-1^ are cell types that are very rare in the larval brain, and hence, could stochastically be lost during sample processing from the starved larvae, and (2) that cells from the UN-2 and neurons^-1^ clusters are indeed present in the data, but due to a low read depth from those cells, have been assigned to other clusters that are present in the starvation condition?

We appreciate that the reviewers brought this point to our attention. Indeed, we cannot exclude the possibility that these cells could stochastically have been lost during sample processing. We have therefore included this possibility in the respective section (“Starvation effects on larval brain cellular composition”) as well as in the Discussion. Nevertheless, among replicates the loss of these clusters continues to be consistent. However, we reason that the sequencing depth is high enough with a saturation higher than 90%.

5) In the VNC, the neural stem cells that exit quiescence at 15hours ALH are the G2 arrested stem cells. Have the authors considered looking at their neural stem cell cluster in the fed vs starved conditions to see if they can distinguish G0 and G2 arrested stem cells, and the effect of starvation on them?

This is a very interesting suggestion and we addressed this *in-silico* and *in-vivo*. We have now incorporated an analysis for the NPCs cell cycle. In additions, we have performed EdU incorporation experiments to demonstrate the reactivation of a subset of neuroblasts prior to 16 hours ALH. Moreover, we have also analyzed the effects of starvation on the NPCs cell-cycle scores upon starvation. This data has been included in the result section and can be now visualized in Figure 2 and Figure 7—figure supplement 3.

*6) Are there any molecular markers in the UN-2 and neurons^-1^ clusters, that the authors could use* in vivo *to demonstrate the loss of those cell types upon starvation?*

This is a very important and challenging point. Neurons^-1^ (now labeled as Neurons X) represents a cluster of neurons expressing a combination of different neurotransmitters, however we could not identify any unique or cluster-specific marker. Similarly, it remains challenging to distinguish UN2s from “classical” neurons.

However, we performed an experiment supporting the loss of UN2s:

We found that *hdc*, was preferentially expressed in UNs in comparison to the remaining cell-clusters, which we confirmed using anti-Hdc immunostaining. Thus, we quantified its expression in normal and starved brains. We found a decreased number of *hdc* positive UNs, a result that supports our *in-silico* observations. Nevertheless, we could only assed this for a combination of UNs1 and UNs2, as for either cluster cell-type specific markers remain unknown. These results were added to the corresponding section (“Starvation effects on larval brain cellular composition”) and are shown in Figure 7—figure supplement 2.

7) A few aspects in the description of the methodology is lacking in detail in order to be able to reproduce the analyses in the paper for future readers. Specifically:

We thank the reviewers for these suggestions and have incorporated the required changes to the manuscript. As this point consists of several subpoints, we have addressed each of them individually.

– Results section “Altogether, our datasets displayed a median of 764,000 reads and1500 genes per cell, and in total of about 12,000 of genes were detected”; Is this the total number of reads, or the number of mapped reads?– Paragraph three of subsection “Single-Cell RNA-Seq of the Drosophila larval brain reveals different categories of characteristic cell-types”: Can the authors state how many cells (what proportion) finally fell into non-neuronal clusters?

We apologize for the missing information and have now added a supplementary table with all the sequencing metrics (Supplementary file 1). Regarding the proportion of cells, approximately 60% of the cells were classified in neuronal clusters, the remaining 40% were assigned to glial cell, NPCs, UNs and nonneural cells. In addition, we have now incorporated a table with the number of cells per cluster (Figure 1—source data 1) and per condition (Figure 1—source data 1).

– Paragraph two of subsection “Mapping neural progenitor cell diversity reveals presence of characteristic neurogenic cell-types”: How was the similarity of expression profiles quantified? Was it in terms of which genes had non-zero expression? "…examined the top differentially expressed genes…"? Is this the top 10 differentially expressed genes, or a different number than 10?

The annotation of each cluster was possible by examining the list of differentially expressed genes, output from “FindAllMarkers” Seurat function. Later, the top 10 genes for each cluster were analyzed to finally confer cluster-identities. We have now modified Figure 1—source data 1 to contain information about the top 10 differentially expressed genes for each cluster, number of cells per cluster and example references that were used to confer cell identity. This information can also be found in the Materials and methods section.

– Subsection “Seurat data processing”: Can the authors describe how many genes and cells were finally considered for analyses in the paper after the filtering criteria mentioned were imposed? Or are the numbers described in the first paragraph of the Results section obtained after the filtering criteria are imposed?

We apologize for not being clear. We have now modified this section and we include both, the raw and filtered number of cells, as well as the total number of genes detected. We first introduced the raw numbers and then described the final number of cells kept for downstream analysis.

“These cells were filtered, scaled and normalized… The resulting 4349 cells, with 12,942 genes detected, were later clustered generating…”

– Subsection “Seurat data processing”: The sequence of operations carried out is unclear. Was the PCA calculated by taking all genes in each cell into account, or only based on the 2000 most highly variable genes? If it was the latter, could the authors explain why the PCA wasn't done with all genes wasn't performed, and add this to the Materials and methods section? On the other hand, if the PCA was calculated with the expression of all genes in each cell, could they explain why the set of the 2000 most variable genes was constructed? Was this set of variable genes used for identifying marker genes or for clustering?

We clarified the processing pipeline by introducing a more detailed description in the Materials and methods section. In brief, 2000 genes were selected based on the recommendations of the R package provider, in order to assess real biological features. These 2000 genes were later considered for downstream processing and analysis.

– Subsection “Seurat data processing”: What is the linear transformation that was applied before PCA was carried out? Why was this linear transformation needed?

We apologize for the brief description and we have now extended the pipeline description this in the Materials and methods section as follows:

“First, a log-normalization with a scale factor of 10,000 was applied to normalize gene expression of individual cells by the total gene expression of each dataset. Second, a linear transformation was executed to remove unwanted source of variation”

– In the same section it appears that three different methodologies were considered for choosing the number of principal components. Could the authors mention which methodology was finally followed for choosing the number of principal components? It will be useful to add these plots as a supplemental figure. Alongside this, could the authors also explain the amount of variance captured each time a given number of principal components is chosen?

This is valid point and it is known the complexity of the process of determining the real dimensionality of a single-cell RNA-seq dataset. Therefore, we have described how we finally determined the number of relevant PCs:

“To define the true dimensionality of the dataset, several approaches were considered: Elbow-Plot and JackStraw-Plot tests together with an evaluation of PC-heatmaps. Finally, principal components (PCs) were selected visually by carefully inspecting Elbow-Plots in order to assess the percentage of variance explained by each PC. In this way, 31 PCs were considered to identify clusters of cells on graph-based approach. To better resolve clusters, a resolution equal to 2 was considered, this parameter helps to subdivide existing clusters and increase the granularity.”

In addition, we have also incorporated a new supplement file containing different Elbow-plots in order to facilitate this process to other researchers.

– UMAP resolution – could the authors add a couple of sentences explaining the impact of increasing or decreasing the UMAP resolution used for clustering? Is this supposed to aid in sub-clustering?

This is correct and we apologize for not describing it previously. This information can be now found in the Materials and methods section:

“In this way, 31 PCs were considered to identify cell clusters with a graph-based approach. To better resolve clusters, we modified the resolution, as increasing this parameter helps to subdivide existing clusters and gain granularity. Therefore, we used a resolution equal to 2 based on the total number of cells and after carefully inspecting different values.”

– Finally, in the interest of reproducibility, it would be nice to make available (through a URL) the scripts that were used to carry out the analyses in the manuscript.

This is indeed a great suggestion and we now provided the scripts used for the analysis as supplementary information.

8) Cell numbers: If we go with their estimate of 2,000 cells per brain, and they analyse 3500 cells (after filtering), then they are sampling the brain with less than a two-fold coverage. Given the tremendous diversity in the Drosophila CNS it would seem that this minimal coverage risks missing a whole range of rare cell types. And this of course has major bearing on the comparison to the starvation conditions as well, which was also less than two-fold. Granted, they do identify some rare cells, such as PDF neurons, but nevertheless, a less than two-fold coverage of an extremely diverse system risks resulting in misinterpretations, especially when comparing between conditions.

This is indeed a valid point and we have performed new single-cell RNA-seq experiments for each condition, adding new biological replicates to each dataset. Now, we have a cell atlas of the first instar larval brain consisting of 9353 cells, representing an approximate 5x coverage of the larval brain.

9) 4-hour starvation seems very short for getting any major transcriptomic changes, and indeed the changes appear to be minor. Is there any evidence for any physiological effect of a 4-hour starvation? Why was a 4-hour starvation chosen? Previously published larval nutrient restriction schemes have been e.g., 24-36 hours (e.g., PMID 21346761). As it now stands, it is difficult to relate the transcriptional changes to any physiological effect(s), and this brings down the impact of the study. Any progress in this area would have greatly strengthened the study.

This is a very relevant point and we thank the reviewers for pointing this out. We have introduced a more detailed description explaining why we decided on the 4 hours starvation.

Since first instar larvae are particularly susceptible to starvation, we could not extend the starvation time. To illustrate this, we have included an experiment showing that after 4 hours of starvation 45.2% of the larvae died. This can be visualized in Figure 7—figure supplement 1. Nevertheless, we strongly agree with the reviewers and we are considering analysis different time points for future experiments, most likely moving to third instar larvae.

10) Related to this issue, studies by the Alex Gould and Andrea Brand labs have revolved around the role of starvation in gating how/when neural progenitors emergence from quiescence during larval stages. This has revealed input from InR, PI3K, Akt, Alk/Jeb and TOR signalling, and some unknown signals, in part provided from the periphery, possibly relayed via CNS surface glia and onto progenitors. The authors of the current study, intriguingly, do find TOR signalling components altered, but do not relate this to the previous studies from Brand and Gould. In fact, somewhat disconcertingly, none of these seminal publications, in Nature, Cell, CellRep, are even mentioned in the current study (PMIDs: 21346761, 21816278, 23478023, 21183078). By analysing responses to starvation separately in glia versus progenitors, the authors have a golden opportunity to shed further light on this intriguing issue, but do not even appear to be aware of the literature in this field.

We thank the reviewers for this indeed valid point. We have now included the mentioned references in our analysis. On the other hand, by analyzing the population of glial cells we observed changes in fat acid metabolism, which can be observed in Figure 8A.

[Editors' note: further revisions were requested prior to acceptance, as described below.]

The manuscript has been improved but there are some remaining issues that need to be addressed before acceptance, as outlined below:1) In the revised version of this manuscript, the authors have addressed and satisfied most of the reviewers' concerns. They have clarified and explained most of the text that made interpretation of their data difficult. They have also added the second replicate of their starved condition. In light of these changes, their work is both interesting and valuable to the field and is recommended for publication.A couple of important points comment, though. There might be an error in their supplementary xls sheet for marker genes. The log-fold changes and p-values associated with some genes in the xls sheet do not seem plausible: there are astronomically high log-fold changes and p-values that are greater than one. For example, consider genes #6,#8, #9 from the "cholinergic neurons 1" cluster. These genes have log-fold-changes of 129,541,248,584,619, 0.616 and 107,496,381,242,778, with associated adjusted p-values of 8.63e-35, 3.49e-17 and 6.24e07.It's possible that these large values occur partly due to the transformations applied to the read counts of the genes in each cell.P-values – adjusted or otherwise – cannot exceed 1, but the p-value of gene #9 is 62.4 million.One might expect p-values to decrease as log-fold-changes increase. So, gene #8, with a fold-change of 0.616, would be expected to have a larger (less significant) p-value than gene #9, which has a fold-change of 107,496,381,242,778. But instead, we see that gene #6 has a p-value of 3.49e-17 while gene #9 has an implausible p-value of 62.4 million.

Thank you for pointing this out. Indeed, we missed that some values were unexpectedly altered after transforming an R output file into a csv file in the previous version of “Figure 1—source data 1”. We have now corrected the error and carefully inspected the numbers in the updated “Figure 1—source data 1”. We apologize for the mistake.

2) Introduction section: "nutriments" should be "nutrients"

This is a valid point and we have now corrected this typo in the manuscript.

3) Results section paragraph one: I find the mean reads/cell to be very high. From our own experience, and from numerous papers, you typically detect 25,000-50,000 gene reads/cell, resulting in the detection of 1000-2500 genes/cell. They list "836,393 mean reads per cell".

Indeed, since the larval brain has fewer cells than the adult counterpart we decided to sequence deeper than it was previously done for the adult brain, we therefore expected a higher number of reads per cell. In the supplementary file we detailed this information for each experiment.